# Massive scalar clouds and black hole spacetimes in Gauss-Bonnet gravity

**Iris van Gemeren[1]⋆, Tanja Hinderer[2]† and Stefan Vandoren[3]‡**

**1** Institute for Theoretical Physics, Utrecht University, Princetonplein 5, 3584 CC Utrecht, The Netherlands

⋆ i.r.vangemeren@uu.nl , † t.p.hinderer@uu.nl , ‡ s.j.g.vandoren@uu.nl

## Abstract

We study static black holes in scalar-Gauss-Bonnet (sGB) gravity with a massive scalar field as an example of higher curvature gravity. The scalar mass introduces an additional scale and leads to a strong suppression of the scalar field beyond its Compton wavelength. We numerically compute sGB black hole spacetimes and scalar configurations and also compare with perturbative results for small couplings, where we focus on a dilatonic coupling function. We analyze the constraints on the parameters from requiring the curvature singularity to be located inside the black hole horizon $r_h$ and the relation to the regularity condition for the scalar field. For scalar field masses $m r_h \gtrsim 10^{-1}$, this leads to a new and currently most stringent bound on sGB coupling constant $\alpha$ of $\alpha/r_h^2 \sim 10^{-1}$ in the context of stellar mass black holes. Lastly, we look at several properties of the black hole configurations relevant for further work on observational consequences, including the scalar monopole charge, Arnowitt–Deser–Misner mass, curvature invariants and the frequencies of the innermost stable circular orbit and light ring.

## 1   Introduction

General Relativity (GR) as the theory of gravity has passed all empirical tests to date [1–3]. Yet modern theoretical developments suggest that modifications of Einstein's gravity are required at some level. This has motivated a significant research effort in high-energy physics to develop a theory of quantum gravity. However, modifications to GR may already arise at intermediate, lower energy scales than the full quantum-gravity regimes. Such modifications have been constrained by high-precision tests of gravity in tabletop experiments [4], the solar system [1], and binary pulsars [5]. However, the genuinely nonlinear regimes of gravity remain largely unexplored and have only recently started to become accessible to measurements, for instance, with gravitational waves [6–8]. This opens new opportunities to test modified theories where corrections to GR only become relevant in high-curvature regimes. One such family of theories is scalar-Gauss-Bonnet (sGB) gravity, where the gravitational action of GR is augmented by adding a quadratic-in-curvature contribution involving the topological Gauss-Bonnet invariant dynamically coupled with a scalar field. Because of the topological nature of the higher curvature term, the theory is ghost-free and the equations of motion are still second order in the fields [9] and thus a dynamical system whose mathematical well-posedness was proved in [10–12]. The sGB form of the gravitational action also has motivations from the low energy limit of quantum gravity paradigms [13–15].

In this paper, we focus on consequences of sGB gravity for static spherical symmetric black holes when including a nonvanishing scalar field mass. Black holes are clean testbeds for precision tests of higher-curvature gravity as they are devoid of any matter and solely involve curved spacetime. In GR, black holes are conjectured to have 'no-hair': their exterior spacetime can be entirely described by only three parameters: their mass, spin, and electromag-

netic charge [16–20]. This also implies that black holes cannot be dressed with any nontrivial scalar, vector, or spinor fields [20–24], even when considering more complex potentials for the fields [25]. The no-hair property of black holes also extends to several classes of modified gravity theories such as Brans-Dicke theories [26] and more generalized scalar-tensor theories [27]. Yet for many other classes of theories, including sGB gravity, the no-hair properties no longer hold. Instead, depending on the parameters, the scalar field can develop a nontrivial profile around black holes that extends through the horizon [12, 28–36] or spontaneous (de-)scalarization can arise [37–42], see the review articles [43, 44] for a detailed discussion.

The scalarization of black holes in sGB strongly depends on properties of the coupling function $f(\varphi)$ between the scalar field and the quadratic curvature terms. When $f(\varphi)$ has a non-vanishing first derivative for all values of $\varphi$, which is often referred to as type I and includes dilatonic couplings $f(\varphi) \sim e^{\gamma\varphi}$, with $\gamma$ a numerical coefficient [28, 29, 45] and linear functions $f(\varphi) \sim \varphi$ leading to shift-symmetric sGB theories [30], only scalarized black hole solutions exist. Studies showed explicitly that black holes evade the no-hair theorem [28, 33, 34] and obtained static [46–48], slowly rotating [29, 49, 50] and rapidly rotating [51–54] black hole solutions. They found that requiring regularity of the scalar field at the horizon leads to an analytical bound in the parameter space beyond which no physical solutions exist [28]. Additionally, the resulting sGB black hole solutions generally have a curvature singularity at a finite radius [47, 55]. For a fixed sGB coupling and smaller black hole masses, the singularity moves farther away from the origin and closer to the horizon. Requiring the absence of naked singularities thus leads to a minimum mass for the domain of existence of black holes. For type II coupling functions whose derivative vanishes for some values of $\varphi$, such as quadratic $f(\varphi) \sim \varphi^2$ [37, 40, 41] and Gaussian $f(\varphi) \sim e^{\gamma\varphi^2}$ [56, 57] couplings, the quadratic scalar field term acts as an effective scalar mass. As the effective mass term can be negative, the black hole solution can become unstable and the presence of scalar condensates becomes favored and results in scalarized black holes.

While black holes in sGB theories with a massless scalar field have been extensively studied as discussed above, the effects of including a scalar field mass remain less explored. Including a mass term in the action is natural from a theoretical perspective and represents the lowest order self-interaction. Accounting for a mass of the scalar field is further motivated by the only scalar field measured to date, the Higgs boson, and common in scalar models for other sectors of particle physics such as the proposed QCD axion and ultralight dark matter candidates [58–62]. A mass term leads to an exponential suppression of effects of the scalar field at scales larger than its Compton wavelength instead of having an infinite extent as in the massless case.

The phenomenology of massive scalar fields around compact objects has been considered in several contexts, including studies of charged black holes [63, 64], black hole superradiance [65–67], neutron stars in scalar tensor gravity [68, 69], and type II sGB black holes [70]. For black holes in type I sGB with a massive scalar field, previous work has numerically calculated black hole solutions [71], included a scalar potential and cosmological constant [72], and studied the dynamics of a massive scalar field with self- interaction in the decoupling limit, i.e. on a fixed Schwarzschild spacetime, via a numerical relativity code [73]. Observational consequences of a massive scalar field in the context of compact objects have also been considered. While the exponential suppression of the field at large distances reduces the size of several of the observational signatures compared to the massless case it may also lead to novel features due to the additional scale involved, as found for gravitational waves from superradiant ultralight boson clouds [74]. Several previous studies further showed that gravitational waves are promising probes for detecting or setting stringent constraints on theories involving massive scalar fields based on effects of scalar dipolar radiation losses in compact-object binary systems. For example, [75] considered binary neutron stars in scalar-tensor gravity, [76]

analyzed extreme mass ratio inspirals, [77] analyzed probing massive fields in the context of multiband detection, and [78] placed the first empirical gravitational-wave constraints on massive sGB.

In this paper, we go beyond previous work on static black holes in massive sGB [71–73] by (i) combining perturbative and numerical analyses to gain deeper insights into the behavior of the spacetime and scalar field and (ii) performing a systematic study of the solutions and resulting observables over a wide parameter space. This differs from the scope of the work in [71], which developed details of the theoretical framework and performed systematic numerical studies of solutions focused on extracting the horizon radius and consequences for thermodynamics. Specifically, in this paper, we numerically compute black hole solutions and, for the first time, also calculate perturbative solutions for small sGB couplings to trace behaviors of the metric functions and scalar field configurations. Together, these two methods enable us to study features of curvature invariants of the spacetime and its energetics such as the gravitational mass and scalar-induced energy density of the configurations from different perspectives. We also analyze the parameter dependencies of the bounds on maximum scalar field at the horizon based on requiring the absence of naked singularities, as obtained from numerical solutions, and regularity of the scalar field at the horizon, as obtained from an analytical bound. This lead to a theory bound on the coupling constant of the gravitational theory. In addition, we calculate the parameter dependencies of observables such as the shifts in the ISCO and light ring away from the GR values. We discuss the relevance of our results as a first step towards making connections with measurements such as the black hole shadows, tidal effects close to the black holes, and as a baseline for computing gravitational wave imprints beyond the leading-order dipole radiation losses. The latter would contribute to the recent ongoing efforts of constructing the gravitational waveforms for black hole binary systems in sGB gravity [79–81]. Our findings also identify interesting mass ranges for the sGB scalar condensate within the broader context of proposed scalar fields in the universe, and highlight interesting qualitative characteristics and parameter ranges for further studies.
In this paper we use Greek indices to denote tensor components in standard Einstein notation. However we use Latin superscripts to assign orders in the small coupling expansion.

## 2  Black holes in scalar-Gauss-Bonnet Gravity

### 2.1  Action

We consider the following action for sGB gravity[1]

$$S_{sGB} = \frac{c^4}{16\pi G} \int_M d^4x \sqrt{-g} [R - 2g^{\mu\nu}\partial_\mu\varphi\,\partial_\nu\varphi - V(\varphi) + \alpha f(\varphi)\mathcal{R}^2_{GB}] \,. \tag{1}$$

Here $R$ denotes the Ricci scalar on manifold $M$ with metric $g_{\mu\nu}$. The scalar field $\varphi$ has potential $V(\varphi)$ and is non-minimally coupled to the Gauss-Bonnet invariant

$$\mathcal{R}^2_{GB} = R^2 - 4R^{\mu\nu}R_{\mu\nu} + R^{\mu\nu\rho\sigma}R_{\mu\nu\rho\sigma} \,. \tag{2}$$

---

[1]For the numerical prefactor of the kinetic and potential (3) scalar field terms, we follow the standard convention also considered for massless sGB, see e.g. [46,79]. However there is a discrepancy in how these factors are defined in the literature on the massive scalar field extension, specifically between [72] and [71]. We follow here the convention of [71], which means that our results of the field equations, metric and scalar field solutions will differ in numerical factors from [72].

via a dimensionless coupling function $f(\varphi)$ and a coupling constant $\alpha$ with dimension length squared. In this work we focus on the simplest potential for a massive scalar field

$$V(\varphi) = 2m^2\varphi^2 \, , \tag{3}$$

where

$$m = \frac{m_\varphi c}{\hbar} \, , \tag{4}$$

denotes the scalar field mass parameter having the dimension of inverse length with $m_\varphi$ the scalar field mass in kilograms. While much of our analysis is general for any coupling function $f(\varphi)$, our case studies of static black hole solutions specialize to type I coupling functions of the form $f(\varphi) = \beta e^{\gamma\varphi}$. This choice is inspired by the low-energy effective action of certain string theories, with the choice of $\beta$ and $\gamma$ corresponding to different string models [82–84]. We will focus here on $f(\varphi) = \frac{1}{4}e^{2\varphi}$ corresponding to the convention for Einstein-dilaton-Gauss-Bonnet (EdGB) gravity [28, 29, 46]. Other choices for $\gamma$ will lead to qualitatively the same behavior for black hole spacetimes [71].

For the massless scalar field theory with this dilatonic coupling, the strongest current observational constraints on the coupling constant $\alpha$ come from a Bayesian analysis of the data from the first three observing runs from the LIGO-Virgo-KAGRA (LVK) detector network to $\sqrt{\alpha} \lesssim 0.8 - 1.33$km [85–87]. For massive scalar field sGB, a first observational constraint based on data from the first two observing runs of LVK obtained $\sqrt{\alpha} \lesssim 2.47$km [78]. A weaker bound in the massive case is consistent with expectations, as the mass causes a suppression of the scalar field effect on large scales.

## 2.2 Relevant length scales

Before discussing the technical details of computing static black hole solutions in massive sGB, we give an overview of the key length scales and their hierarchy, which has important consequences for qualitative features of the solutions and for defining perturbative approximations. Figure 1 illustrates these scales for an example of a black hole and scalar condensate. We consider a static, spherically symmetric black hole of horizon radius $r_h$ which is of the order (but slightly smaller [71]) of the Schwarzschild radius

$$r_h \sim r_S = \frac{2GM}{c^2} \, , \tag{5}$$

with $M$ the mass of the black hole. The black hole is surrounded by a massive scalar field cloud that extends inside the horizon. The characteristic size of the cloud is related to the mass of the scalar field. The cloud is exponentially suppressed for distances beyond the Compton wavelength $\lambda_\varphi$ which is inversely proportional to the scalar field mass $m$

$$\lambda_\varphi \sim 1/m \, . \tag{6}$$

Hence in the small-mass limit the scalar field cloud stretches out further to infinity, approaching the massless sGB solution. By contrast, for larger masses, the scalar field becomes more confined to the vicinity of the horizon, and for $m \to \infty$ the scalar field decouples and the solution approaches the Schwarzschild black hole. In Fig. 1 we show the Compton wavelength length scale for small masses. Here, small masses refers to the Compton wavelength being larger than the black hole horizon.

The last length scale is set by the coupling constant $\sqrt{\alpha}$ which determines the strength of the higher curvature contributions. When we apply perturbation theory in section 3, we assume the dimensionless version of the coupling to be small

$$\hat{\alpha} \equiv \frac{\alpha}{r_h^2} \, . \tag{7}$$

Assuming the current observational bound is saturated $\sqrt{\alpha} = 2.47$ km and considering black holes in the mass range $5M_\odot \lesssim M \lesssim 10^{10}M_\odot$, the dimensionless coupling lies in the range $10^{-11} \lesssim \hat{\alpha} \lesssim 0.2$, validating the assumption of working in the small coupling regime. The perturbation theory we set up is exact in $m$, i.e. we do not assume any restriction on the scalar field mass. Expanding both in the small mass and coupling limit resulted in non-regular solutions for the scalar field at the black hole horizon. On the other hand, when discussing the numerical solution to the field equations, no restrictions on the length scales related to both the mass and coupling are assumed. However it turns out that requiring the scalar field to be regular at the horizon does give a restriction on the value of the coupling and scalar field mass depending on the black hole mass and amount of scalar field at the horizon. This restrictions ensures that the curvature singularity at $r \neq 0^2$ lies within the horizon and hence prevents a naked singularity, see also Fig.1.

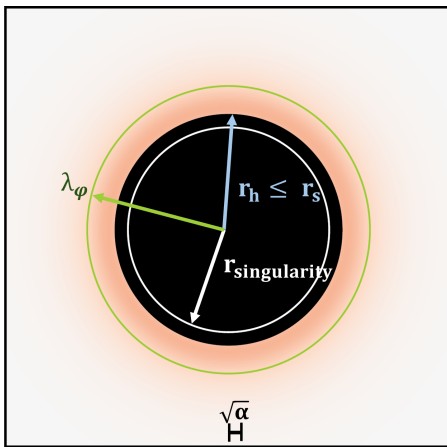

Figure 1: Sketch of the black hole horizon (black region) and the scalar condensate around the black hole (red) with the relevant length scales and their hierarchy in an example of a small scalar field mass.

## 2.3    Field equations

Varying the action (1) with respect to the metric $g_{\mu\nu}$ results in the following field equations

$$G_{\mu\nu} = T_{\mu\nu} \,, \tag{8}$$

with $G_{\mu\nu}$ the Einstein tensor and $T_{\mu\nu}$ the 'effective' energy momentum tensor which includes contributions from the scalar field and the higher curvature terms,

$$T_{\mu\nu} = 2\partial_\mu\varphi\,\partial_\nu\varphi - g_{\mu\nu}\partial_\rho\varphi\,\partial^\rho\varphi - g_{\mu\nu}m^2\varphi^2 - 4\alpha\,^*R^*_{\alpha\mu\nu\beta}\nabla^\alpha\nabla^\beta f(\varphi) \,. \tag{9}$$

Here $^*R^*_{\alpha\mu\nu\beta}$ is the double dual Riemann tensor defined as $^*R^*_{\alpha\mu\nu\beta} = \frac{1}{4}\epsilon_{\alpha\mu}{}^{\gamma\sigma}R_{\gamma\sigma\rho\varepsilon}\epsilon^{\rho\varepsilon}{}_{\nu\beta}$ with $\epsilon_{\alpha\mu\gamma\sigma}$ the anti-symmetric Levi-Civita tensor. The scalar field equation is given by

$$\Box\varphi = m^2\varphi - \frac{1}{4}\alpha f'(\varphi)\mathcal{R}^2_{\text{GB}} \,, \tag{10}$$

with $\Box \equiv g^{\alpha\beta}\nabla_\alpha\nabla_\beta$ the d'Alembertian operator. One can check that the field equations (8), (10) are invariant under the rescaling of the coordinates with a generic factor $c$ which leaves the fields invariant together with redefining $m \to m/c$, $\alpha \to c^2\alpha$.

---

[2] When working in Schwarzschild coordinates.

### 2.4  Metric and asymptotic behavior

In this work we focus on static, spherically symmetric black hole solutions for which the general metric is given by

$$ds^2 = -e^{A(r)}dt^2 + e^{B(r)}dr^2 + r^2(d\theta^2 + \sin^2\theta\, d\phi^2). \tag{11}$$

We assume the same symmetries for the scalar field, hence $\varphi = \varphi(r)$. Substituting this and the general metric (11) in the field equations (8) and (10), we obtain the components of (8) given explicitly in (47) in Appendix A. The scalar field equation is given explicitly by (48). To obtain the desired black hole and condensate solutions to the equations of motion for the metric functions $A(r)$, $B(r)$ and the scalar field $\varphi(r)$ requires imposing the correct boundary conditions at the black hole horizon and at spatial infinity. The black hole horizon is defined in Schwarzschild coordinates by a vanishing time component of the metric and a diverging radial component. Furthermore we require the scalar field to remain regular at the horizon. Hence we have the following conditions approaching the black hole horizon $r_h$

$$\begin{aligned}
A(r) &\to -\infty\,,\\
B(r) &\to \infty\,,\\
\varphi'(r), \varphi''(r) &\,\text{finite}\,.
\end{aligned} \tag{12}$$

Furthermore at infinite radial distance we require the solution to be asymptotically flat and approach Minkowski spacetime. Therefore, at spatial infinity, the scalar field sourcing the metric equations should vanish as well and we have

$$\begin{aligned}
A(r) &\to 0\,,\\
B(r) &\to 0\,,\\
\varphi(r) &\to 0\,.
\end{aligned} \tag{13}$$

To capture the nontrivial fall-off behavior of the scalar field near infinity, we substitute the asymptotic metric functions (13) in the scalar field equation (48) and obtain

$$2r^2\varphi''(r) + 4r\varphi'(r) - 2m^2 r^2 \varphi(r) = 0\,. \tag{14}$$

Solving this differential equation for $\varphi(r)$ yields the asymptotic solution

$$\varphi(r) \to c_1 \frac{e^{-mr}}{r} + c_2 \frac{e^{mr}}{2mr}\,, \tag{15}$$

with $c_1$, $c_2$ two integration constants. For an asymptotically flat solution we require $c_2 = 0$ while the remaining coefficient $c_1$ is determined by matching to the near-horizon solutions and depends on the coupling as we show in Sec. 6.3.2. The expression (15) with $c_2 = 0$ quantifies the qualitative behavior alluded to earlier: the scalar field mass causes the field configuration to be constrained to the vicinity of the black hole and exponentially suppressed beyond the scale of the Compton wavelength (6). In the limit $m \to 0$, the exponential in (15) becomes unity and the falloff of the field is much slower $\sim 1/r$, consistent with calculations in the massless case [46,47].

## 3  Perturbative black hole solutions for small coupling

Before we compute the exact metric and scalar field solutions by solving the field equations (8), (10) numerically, we analyze the solution in the small coupling expansion to gain further insights

into the behavior of the solution. We expand in the dimensionless coupling constant $\hat{\alpha}$ defined in (7). It is convenient to define a dimensionless radial coordinate

$$u = \frac{r_h}{r} \ , \tag{16}$$

so the horizon always lies at $u = 1$ and spatial infinity at $u = 0$. Furthermore we introduce the dimensionless mass

$$\hat{m} = r_h m \ . \tag{17}$$

We expand the metric components and the scalar field for small coupling $\hat{\alpha} \ll 1$. At this stage it is more convenient to reparameterize the metric functions

$$
\begin{aligned}
e^{A(u)} &\to \bar{A}(u) \ , \\
e^{B(u)} &\to \frac{1}{\bar{B}(u)} \ ,
\end{aligned}
\tag{18}
$$

as it makes the expansion more straightforward. Then the small-coupling expansion is given by the ansatz

$$
\begin{aligned}
\bar{A} &= \sum_{i=0}^{\infty} \bar{A}^i \, \hat{\alpha}^i \ , \\
\varphi &= \sum_{i=0}^{\infty} \varphi^i \, \hat{\alpha}^i \ ,
\end{aligned}
\tag{19}
$$

where we omit here and in the following the explicit expansion of $\bar{B}$ as it is similar to (19). We substitute this ansatz into (47), (48) and solve order by order in $\hat{\alpha}$. At $\mathcal{O}(\hat{\alpha}^0)$ we need to obtain the Schwarzschild solution as the limit of $\alpha \to 0$ should recover GR. Therefore we can already impose

$$
\begin{aligned}
\bar{A}^0 &= \bar{B}^0 = 1 - u \ , \\
\varphi^0 &= 0 \ .
\end{aligned}
\tag{20}
$$

To recover the Schwarzschild solution at zeroth order in the coupling, in the context of the perturbative solution $r_h$ in (16), (17) and (7) is equal to $r_S$ (5). However we defined the variable $u$, mass and coupling parameters in terms of the general horizon radius so they can be used in the context of the exact solution in Sec. 4 as well.

## 3.1 Equations of motion at linear order in the coupling

Before analyzing in detail the expansion of the field equations, we can already gain insights into the scalings of different contributions with the coupling by considering the field equations (47) with the expansion (19). At linear order in the coupling, there is a correction to the scalar field as the source term in (10) is linear in the coupling. Next, analyzing the source of the metric equations of motion (9) we find that the energy momentum tensor consists of terms quadratic in the scalar field and a contribution linear in the coupling times $\nabla^\alpha \nabla^\beta f(\varphi) = \nabla^\alpha (f'(\varphi) \partial^\beta \varphi)$ which is at least linear in the scalar field. As the scalar field to lowest order is linear in the coupling, $T_{\mu\nu}$ is quadratic and higher order in $\hat{\alpha}$. Consequently, the corrections to the field equations for the metric potentials (47) will only appear $O(\hat{\alpha}^2)$. At linear order in $\hat{\alpha}$, the metric remains the Schwarzschild metric and we need to compute the solution to the linearized scalar field equation in a Schwarzschild background. This is summarized in the second row of Table 1. In particular, to solve for the linear solutions in $\hat{\alpha}$, we substitute the small coupling expansion for the scalar field and metric components (19)

258  in (48) and use (20) for the zeroth order coefficients and $\bar{A}^1 = \bar{B}^1 = 0$ as discussed above.
259  This leads to the linearized scalar field equation in the radial coordinate $u$ defined in (16)

$$(u-1)\varphi^{1\prime\prime}(u) + \varphi^{1\prime}(u) + \frac{\hat{m}^2}{u^4}\varphi^1(u) = 3u^2 f'(\varphi^0) \,. \tag{21}$$

### 3.1.1  Near-horizon and asymptotic behavior of the linearized field

261  To capture the solution of (21) near the horizon, we expand around the horizon radius

$$\epsilon = u - 1 \,. \tag{22}$$

262  This leads to a double expansion of the fields in $\hat{\alpha}$ and $\epsilon$, where each coefficient in the $\hat{\alpha}$
263  expansion in (19) is further expanded in a Taylor series in $\epsilon$. For the $O(\hat{\alpha})$ coefficient we have

$$\varphi^1 = \varphi_h^1 + \epsilon \varphi_h^{1\prime} + \mathcal{O}(\epsilon^2) \,. \tag{23}$$

264  For the $O(\hat{\alpha})$ terms, solving the differential equation (21) order by order in $\epsilon$ and using (20)
265  determines $\varphi_h^{1\prime}$ in terms of $\varphi_h^1$ via

$$\varphi_h^{1\prime} = 3f'(0) - \hat{m}^2 \varphi_h^1 \,. \tag{24}$$

266  The coefficient $\varphi_h^1$ corresponds to the amount of scalar field at the horizon at linear order in
267  the coupling and $f'(0)$ is a constant. One can reason that the solution to (21) has to be a
268  monotonically increasing solution (see Appendix B for the detailed arguments) and therefore
269  the first derivative at the horizon has to be positive [64]. This leads to the following constraint
270  on the amount of (linearized) scalar hair at the horizon and the mass of the scalar field

$$\varphi_h^1 < \frac{3f'(0)}{\hat{m}^2} \,. \tag{25}$$

271  In the linearized case, we thus find a constraint on the amount of scalar field at the horizon.
272  In massless sGB, similar arguments result in an expression for the scalar field derivative at the
273  horizon (in this case for the full theory) [28] given by

$$\varphi_h' = \frac{r_h}{4\alpha f'(\varphi_h)}\left(-1 \pm \sqrt{1 - \frac{24\alpha^2 f'(\varphi_h)^2}{r_h^4}}\right) \,. \tag{26}$$

274  Requiring the square root to be positive yields the constraint

$$f'(\varphi_h)^2 < \frac{r_h^4}{24\alpha^2} \,. \tag{27}$$

275  For a fixed coupling function and constant, this bound (27) determines the maximum amount
276  of allowed scalar hair at the horizon depending on the size of the black hole. Conversely,
277  given a certain amount of scalar field at the horizon, the constraint (27) sets a lower bound
278  on the black hole mass that can sustain this hair. Using the definition of $\hat{m}$ from (17) in (25)
279  shows that the maximum amount of scalar hair at the horizon depends both on the scalar field
280  mass and the black hole mass. In Sec. 6 below we study the effect of the scalar mass on these
281  quantities with full black hole solutions and establish a more meaningful comparison to the
282  massless results (27).

283

284      As at linear order in the coupling the background is still Schwarzschild spacetime, the
285  asymptotic limit of the scalar field at this order follows (15) to first order in the asymptotic
286  expansion in $u$. To write it in the notation introduced in this section

$$\varphi^1(u) = \varphi_\infty^{1\prime} e^{-\hat{m}/u} u + \bar{\varphi}_\infty^{1\prime} \frac{e^{\hat{m}/u}}{2\hat{m}} u + \mathcal{O}(u^2) \,, \tag{28}$$

287  where we absorbed the factor $r_h$ in the first term in the constant $\varphi_\infty^{1\prime}$.

### 3.1.2 Numerical solution for the linearized field

The solution to (21) has to be calculated numerically. It is computed by defining an initial value problem at an infinitesimal distance from the black hole horizon $u = 1 - 10^{-5}$, with (23) as initial condition, and integrating to spatial infinity $u = 0$. We keep the description and discussion of the numerical methods needed on top of a numerical integrator general. For the numerical integration we specify to an 8th order explicit Runge Kutta scheme with a machine and working precision of 30 digits to acquire the needed numerical precision. For more details we refer to the last section of Appendix C.

In (23) $\varphi_h^1$ is the constant that needs to be determined by matching to the asymptotic limit (28). A difficulty is to ensure that the asymptotic solution (28) obeys the desired fall-off conditions at infinity, with $\bar{\varphi}_\infty^{1\prime} = 0$ to eliminate the growing mode. If this condition is not exactly fulfilled, the growing mode always takes over at some large distance from the horizon. Furthermore, any small numerical error in the initial condition that results in an inexact match to $\varphi_\infty^{1\prime}$ finite and $\bar{\varphi}_\infty^{1\prime}$ zero in (28) immediately leads to a diverging solution. Therefore, finding the exact exponentially decaying solution numerically is a challenge. However, solutions close to the desired solution can be computed using the bisection method described in [64] and in Appendix C. This method is based on identifying the domain of existence of the exponentially decaying solution in the range of input guesses $\varphi_h^1$ for which, when integrating the solution outwards, the behavior at infinity switches from positively to negatively diverging for too large or too small guesses respectively. Decreasing this range for $\varphi_h^1$ through several iterations leads to a narrow range of guesses that approach the 'right' value for $\varphi_h^1$ such that the solution only decays. The more cycles in this bisection method, the more accurate the guess for $\varphi_h^1$ and the farther the diverging behavior is pushed out to larger distances. This is shown in Fig. 2 below. For solving (21) we apply this bisection method for 15 cycles, where the difference in $\varphi_h^1$ from

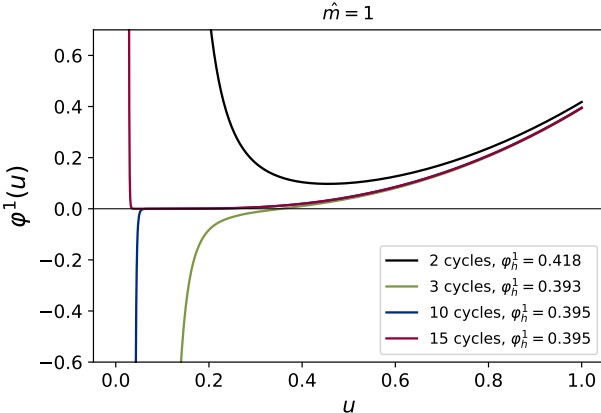

Figure 2: *Solution for the linearized scalar field with $\hat{m} = 1$ for different numbers of cycles of the bisection method.* The legend also shows the value of $\varphi_h^1$ corresponding to each curve. The integration starts at the horizon $u = 1 - 10^{-5}$ and proceeds outwards to infinity $u = 0$.

the value of the previous cycle is $\sim 10^{-14}$. We compare these small-coupling results for $\varphi_h^1$ to the values obtained in the full solution in Sec. 4.

| Order | $tt$ field equation | $rr$ field equation | scalar field equation |
|---|---|---|---|
| $\hat{\alpha}^0$ | $\bar{B}^0, \varphi^0$ | $\bar{A}^0, \bar{B}^0, \varphi^0$ | $\bar{A}^0, \bar{B}^0, \varphi^0$ |
| $\hat{\alpha}^1$ | - | - | $\varphi^1$ |
| $\hat{\alpha}^2$ | $\bar{B}^2, \varphi^1$ | $\bar{A}^2, \bar{B}^2, \varphi^1$ | $\varphi^2$ |
| $\hat{\alpha}^3$ | $\bar{B}^3, \varphi^1, \varphi^2$ | $\bar{A}^3, \bar{B}^3, \varphi^1, \varphi^2$ | $\bar{B}^2, \bar{A}^2, \varphi^1, \varphi^2, \varphi^3$ |

Table 1: *Dependencies of the equations of motion* (47) *and* (48) *in the small-coupling limit on the expansion coefficients at each order* $n$ *in* $\hat{\alpha}$. *At orders* $n > 0$, *the dependencies listed in the table are those obtained after substituting the lower order solutions.*

## 3.2  Higher order corrections in $\hat{\alpha}$

As discussed in Sec. 3.1, the corrections to the Schwarzschild metric first appear at order $\hat{\alpha}^2$. At each $n$-th order in the perturbative expansions in $\hat{\alpha}$ with $n \geq 2$, the field equations (47) together with the background and linearized solutions discussed above depend on the metric coefficients at orders $\leq n$ as well as the scalar field corrections up to one lower order $\leq n-1$. The scalar field equation (48) becomes dependent on the metric corrections only at $O(\hat{\alpha}^3)$. Therefore we focus on obtaining the perturbative solution to that order so as to capture all the different dependencies of the solutions and compare with the full solution in the next section. We summarize these dependencies of the field equations at the different orders in Table 1. The approach to assemble all the inputs to compute solutions is similar to the linearized case: after obtaining the system of equations order by order in $\hat{\alpha}$ from the small-coupling expansion of the field equations, the next step is to analyze their asymptotic and near-horizon limits.

### 3.2.1  Near horizon and asymptotic limit of the higher order correction solutions

For the near horizon limit, we expand all functions $\epsilon$ defined in (22) as in the linearized case 3.1. Specifically, we make the ansatz

$$\bar{A}^i = \bar{A}^i_h + \epsilon \bar{A}^{i\prime}_h + \epsilon^2 \bar{A}^{i\prime\prime}_h + \mathcal{O}(\epsilon^3) \,,$$
$$\varphi^i = \varphi^i_h + \epsilon \varphi^{i\prime}_h + \epsilon^2 \varphi^{i\prime\prime}_h + \mathcal{O}(\epsilon^3) \,, \tag{29}$$

and similarly for $\bar{B}^i$, where we focus on $i = 2, 3$ for the quadratic and cubic orders in the coupling respectively. We substitute this ansatz into the $tt$ and $rr$ components of the field equations (47) and the scalar equation of motion (48), expand for $\epsilon \ll 1$ and solve order by order.

To capture the asymptotic behavior at spatial infinity, we first note that as discussed above, the corrections to the scalar field equation of motion from the metric enter only at $O(\hat{\alpha}^3)$. Thus, at $O(\hat{\alpha}^2)$, the asymptotic behavior of $\varphi^2$ is still given by (15). By contrast, the metric field equations (47) at $O(\hat{\alpha}^2)$ and higher depend on the scalar field one order lower in $\hat{\alpha}$ (see Table 1). Thus, near spatial infinity they involve contributions from a quadratic combination of the scalar field asymptotics (15) with $c_2 \to 0$. In turn, this implies that at $O(\hat{\alpha}^3)$, the asymptotic scalar field involves a cubic combination of (15). Based on these considerations, we include the expected number of factors of the exponential from (15) in our ansatz for the expansion of the functions near spatial infinity, specifically

$$\bar{A}^i = e^{-2\hat{m}/u} \left( \bar{A}^i_\infty + u \bar{A}^{i\prime}_\infty + u^2 \bar{A}^{i\prime\prime}_\infty + u^3 \bar{A}^{i\prime\prime\prime}_\infty + \mathcal{O}(u^4) \right) \,,$$
$$\varphi^3 = e^{-3\hat{m}/u} \left( \varphi^3_\infty + u \varphi^{3\prime}_\infty + u^2 \varphi^{3\prime\prime}_\infty + u^3 \varphi^{3\prime\prime\prime}_\infty + \mathcal{O}(u^4) \right) \,, \tag{30}$$

and similarly for $\bar{B}^i$ again focussing on $i = 2, 3$. With the dependencies on the exponentials captured in the ansatz, one can factor them out in the field equations to the lowest orders

in $u$ (here up to $u^3$). Factoring out the exponentials is important to be able to proceed, as otherwise the field equations do not have a series expansion around $u = 0$ since $e^{1/u}$ remains large in this limit. In (30) we only kept terms up to $O(1/u^3)$, which we found to give sufficient accuracy for our purposes. However, the method can be extended to include higher orders by altering the ansatz in such a way that the dependencies on exponentials can be factored out in the equations of motion.

We substitute the ansatz (30) into the $tt$, $rr$ components of the field equations and the scalar one at each $O(\hat{\alpha}^i)$ and solve order by order in $u$ for the coefficients. We find that, as expected based on the scaling considerations discussed above, these coefficients depend on the scalar field integration constants up to one order lower in $\hat{\alpha}$.

### 3.2.2   Numerical solutions with higher order corrections

With the asymptotics near the horizon and spatial infinity in hand, we turn to solving the field equations over the entire spatial domain order by order in $\hat{\alpha}$. We first note the simplifying fact that at quadratic order in $\hat{\alpha}$, the $tt$ component of the field equations at $O(\hat{\alpha}^2)$ depends only on the $\bar{B}$ correction and the scalar field at $O(\hat{\alpha})$ (see Table 1). We can therefore first solve the $tt$ component of the field equations at $O(\hat{\alpha}^2)$ for $\bar{B}$ by substituting the numerical solution of the linearized scalar field as described in Sec. 3.1 and solving the equation numerically by starting the integration from an infinitesimal distance outside of the horizon $u = 1 - 10^{-5}$ and integrating towards $u = 0$ using the same specifications for the numerical integrator as mentioned in Sec. 3.1.2. As discussed in Sec. 3.1 and Appendix C, the divergent behavior of the linearized solution, which enters into all subsequent calculations at higher orders in $\hat{\alpha}$, can numerically only be suppressed out to a small but finite $u$. This implies that the higher order solutions can only be computed up to a slightly larger value of $u$, as the onset of the divergence must be pushed outside the domain of integration. For a given accuracy of the linearized solution, this leads to a deterioration in accuracy at each higher order in $\hat{\alpha}$.

For the initial conditions of the integration we use (29) to linear order in $\epsilon$. For $\bar{B}^2$ this is given by

$$\bar{B}^2 \sim \bar{A}_h^2 + \epsilon \left( (\hat{m} \varphi_h^1)^2 + \bar{A}_h^2 \right) . \tag{31}$$

The coefficient $\bar{A}_h^2$ needs to be determined by matching to the asymptotic limit (30). The asymptotic solution of $\bar{B}^2$ in (30) is given by

$$\bar{B}^2 \sim e^{\frac{-2\hat{m}}{u}} \left( \hat{m} \varphi_\infty^{1\prime} u + \frac{1}{2} u^2 \left( 2(\varphi_\infty^{1\prime})^2 - \hat{m}(\varphi_\infty^{1\prime})^2 \right) \right) , \tag{32}$$

with $\varphi_\infty^{1\prime}$ the integration constant of the asymptotic limit of the linearized scalar field (28) and is thus completely determined by the scalar field solution at linear order in the coupling. We compute the numerical solution having this desired asymptotic behavior by using a shooting method. This is based on obtaining the solution for $\bar{B}^2$ for different guesses of $\bar{A}_h^2$ and evaluating these solutions at infinity until these values agree with the values at infinity of (32). In Appendix C we describe details of the implementation of the shooting method in this context by giving the explicit example for computing $\bar{B}^2$.

Having solved the $tt$ component of the field equations, we use the resulting numerical solution for $\bar{B}^2$ together with $\varphi^1$ in the $rr$ field equation and solve for $\bar{A}^2$ using the shooting method described above and in Appendix C. This completes the computation of the metric functions at $O(\hat{\alpha}^2)$. The solution for the scalar field expansion coefficient $\varphi^2$ at that order can be determined separately, as its equation of motion does not involve any metric corrections (see Table 1). Therefore we can use the same bisection method as for the linearized scalar field. Finally, the metric and scalar field corrections at $O(\hat{\alpha}^3)$ can be determined via the same procedure and methods as described for the second order corrections.

## 4  Full numerical black hole solutions

To check to what extent the perturbative solution captures the behavior of the black hole spacetime correctly and compute results including non-perturbative effects, we solve the field equations (36) without approximations using numerical methods. We follow the methodology of [71] for a specific choice of coupling function, however, our analysis in Sec. 6 has a different focus and therefore complements the results in [71].

For solving the full field equations, it is more convenient to work with a different setup from that used for the small-coupling approximations described above. In particular, we work with the parameterization of the metric potentials in terms of $A$ and $B$ instead or $\bar{A}$ and $\bar{B}$ and rewrite (47), (48) as follows [47, 72]. We use the $rr$-component to eliminate the $B(r)$ and $B'(r)$ contributions to the field equations and cast the $rr$-component (47) as a quadratic equation in $e^{B(r)}$

$$e^{2B(r)}\rho(r) + e^{B(r)}\beta(r) + \gamma(r) = 0 \,, \tag{33}$$

where

$$\begin{aligned}
\rho(r) &= 4\left(1 - (mr\varphi(r))^2\right) \,, \\
\beta(r) &= -4\left(1 + rA'(r) + 2\alpha A'(r)f'(\varphi)\varphi'(r) - r^2\varphi'(r)^2\right) \,, \\
\gamma(r) &= 24\alpha A'(r)f'(\varphi)\varphi'(r) \,.
\end{aligned} \tag{34}$$

The solution to the quadratic equation (33) is given by

$$e^{B(r)} = \frac{-\beta(r) + \sqrt{\beta(r)^2 - 4\rho(r)\gamma(r)}}{2\rho(r)} \,. \tag{35}$$

Here, we chose the solution with the positive sign as it gives the desired asymptotic limit[3] defined by (13). Furthermore, the expression for $B'(r)$ is given by the derivative of (35). The remaining field equations can then be rewritten as two second order differential equations for $A(r)$ and $\varphi(r)$ given explicitly by

$$\begin{aligned}
A''(r) &= f(r, \varphi(r), \varphi'(r), A'(r)) \,, \\
\varphi''(r) &= h(r, \varphi(r), \varphi'(r), A'(r)) \,.
\end{aligned} \tag{36}$$

Here $f$ and $h$ are functions of the corresponding variables in their arguments, which are given in by (49), (50). We note that in obtaining (36) we focused on rewriting the $\theta\theta$ and scalar field equations (47), (48), however the final solutions of the metric function $A(r)$ and $\varphi$ are independent of this choice. In practice, finding the black hole solution requires solving (36) for $A(r)$ and $\varphi(r)$ as a boundary value problem corresponding to (12) and (13).

### 4.1  Near-horizon and asymptotic behavior of the exact solutions

As our goal to obtain the spherically symmetric black hole solution has been reduced to solving the boundary value problem corresponding to (36), we study in this section the behavior of the metric functions and scalar field approaching these boundaries in more detail following [47, 72].

---

[3]Substituting the asymptotic behavior for $A(r)$ and $\varphi(r)$ assuming both fall off to $0$ as $\sim 1/r$ and $\sim e^{-mr}/r$ respectively , which we discuss in Sec. 4.1.1, leads to $\beta(r) \to -4$. Then the positive sign solution gives $e^{B(r)} \to 1$ which is the desired asymptotically flat result.

### 4.1.1 Asymptotic limit

From our estimate in section 2.4, by substituting in this limit the Minkowski metric in the field equations, we found that the scalar field falls of as $\sim e^{-mr}/r$ to first order in $1/r$. We also limit the expansion of the asymptotic limit for the full solution to first order in the $1/r$. This is motivated by the perturbative results of Sec. 3.2.1, which showed that higher order corrections in $1/r$ occur together with higher order powers of the exponent, hence these corrections are strongly suppressed. For the order $1/r$ correction to the metric functions, we can make the following argument. As the scalar field falls of exponentially, at spatial infinity the scalar field has decreased to zero. In the case of zero scalar field, the higher curvature corrections to the field equations vanish as well, see (47). This can also be reasoned from the action (1), where for a vanishing scalar field, the prefactor of the GB invariant is constant and because the term is a topological invariant it becomes a boundary term and its contribution to the dynamics vanishes. The asymptotic behavior at order $1/r$ of the metric function $e^{A(r)}$ and $e^{B(r)}$ therefore correspond to the Schwarzschild metric. Again we know from the perturbative case that in this regime, higher orders in $1/r$ are strongly suppressed. The asymptotic behavior of the functions in (36) is then given by

$$
\begin{aligned}
e^{A(r)} &\to \frac{A'_\infty}{r} + \mathcal{O}(1/r^2) \,, \\
\varphi(r) &\to \frac{\varphi'_\infty e^{-mr}}{r} + \mathcal{O}(1/r^2) \,.
\end{aligned}
\tag{37}
$$

The integration constants $A'_\infty$ and $\varphi'_\infty$ are proportional to the system's ADM mass and scalar monopole charge respectively and are fixed by matching the solution to the near horizon limit detailed below.

### 4.1.2 Near horizon limit

The behavior of the metric functions and the scalar field at the horizon is given by (12). The divergence in the function $A(r)$ implies $A'(r) \to \infty$. Thus, $1/A'(r) \to 0$ and we expand the field equations (35) in $1/A'(r)$, which leads to

$$
\begin{aligned}
e^{B(r)} = &\frac{2\alpha f'(\varphi)\varphi'(r) + r}{(1 - r^2 m^2 \varphi(r)^2)} A' + \left[ 2\alpha f'(\varphi)\varphi'(r)\left(2 - 3m^2 r^2 \varphi(r)^2 + r^2 \varphi'(r)^2\right)\right. \\
&\left. + r\left(r^2 \varphi'^2 - 1\right)\right] / \left[\left(r^2 m^2 \varphi(r)^2 - 1\right)\left(2\alpha f'(\varphi)\varphi'(r) + r\right)\right] + \mathcal{O}\left(\frac{1}{A'}\right) \,.
\end{aligned}
\tag{38}
$$

Substituting the expanded expression (38) in (36) and expanding the equations in the same limit gives

$$
\begin{aligned}
A''(r) &= \frac{a}{b} A(r)^2 + \mathcal{O}\left(A'\right) \,, &\tag{39a} \\
\varphi''(r) &= \frac{c}{b}\left(2\alpha f'(\varphi)\varphi'(r) + r\right) A'(r) + \mathcal{O}(1) \,, &\tag{39b}
\end{aligned}
$$

where $a$, $b$ and $c$ are given by (51). For $\varphi''(r)$ to remain finite as $A'(r) \to \infty$, we require the coefficient of $A'$ in (39b) to vanish at a rate equal or faster than $A'$ diverges. However, comparing (39) with (38), we see that letting $(2\alpha f'(\varphi)\varphi'(r) + r)$ vanish would also make the divergent term $\sim A'$ of $e^{B(r)}$ vanish, which is inconsistent with the horizon condition (12). Therefore, to impose regularity of the scalar field near the horizon requires $c \to 0$ and $b \neq 0$.

At the black hole horizon we can rewrite $c = 0$ using the explicit expression (51) as a condition on $\varphi'(r_h) = \varphi'_h$ given by

$$
\varphi'_h = -\frac{A \pm \left(1 - m^2 r_h^2 {\varphi_h}^2\right)\sqrt{C}}{B} \,,
\tag{40}
$$

with $A$, $B$ and $C$ given by (52) and (53). Only the minus solution converges to (26) in the small mass limit and to (24) in the small coupling limit. We also note that the square root in (40) adds an additional requirement as it should be positive definite, imposing an inequality which gives a further restriction on the parameters $\varphi_h$, $r_h$.

Next, considering the near-horizon expansion of the field equations and substituting the minus solution of (40) in (39) yields

$$\begin{aligned} A'' &= -(A')^2 + \mathcal{O}\left(A'\right), \\ \varphi'' &= \mathcal{O}(1). \end{aligned} \tag{41}$$

Integrating (41) yields a logarithmic function and fixing the integration constant such that the solution diverges to minus infinity at $r_h$ leads to the derivative $A'(r) \sim \frac{1}{r-r_h}$. Combining this with (38) we obtain the near-horizon behavior of the metric components and scalar field

$$\begin{aligned} e^{A(r)} &= A'_h \left(r - r_h\right) + \mathcal{O}(r - r_h), \\ \varphi(r) &= \varphi_h + \varphi'_h \left(r - r_h\right) + \mathcal{O}(r - r_h), \end{aligned} \tag{42}$$

where $\varphi'_h$ is given by (40). Then $A'_h$, $\varphi_h$ are the only free integration constants which get fixed by matching with the asymptotic solution.

## 4.2 Numerical computation of the full solution

We use an initial value formulation to solve the second order differential equations (36) for $A(r)$ and $\varphi(r)$ simultaneously again using the same specifications for the numerical integrator as mentioned in Sec. 3.1.2. Note that $\alpha$, $m$ and $r_h$ are all input parameters in this initial value problem. The solution for $B(r)$ can be recovered by substituting these solutions in (35). We start the integration at an infinitesimal distance ($r/r_h = 1 + 10^{-3}$) outside the event horizon using the near-horizon solutions (42) and (40) as initial conditions. The amount of scalar field at the horizon $\varphi_h$ and the coefficient $A'_h$ are determined by matching to the right asymptotic behavior. The unstable nature of the scalar field solution poses a challenge for solving (36) simultaneously with the right asymptotic behavior. It turns out that the scalar field solution and approximation for $\varphi_h$ are not sensitive to the estimation for $A'_h$. One can therefore obtain an educated guess for $\varphi_h$ independent of $A'_h$ and use this guess to solve the system simultaneously. The scalar field solution up to some finite value of $r$ then already behaves as the exponentially decaying solution and a numerical root finding routine is then able to extract the initial conditions corresponding to the right asymptotic behaviors.

More explicitly, we implement these considerations as follows. After defining the system of differential equations (36) as functions of the initial values $\varphi_h$, $A'_h$, we use the bisection method described in Sec. 3.1 and Appendix C to obtain an educated guess for $\varphi_h$, setting $A'_h$ temporarily to 1. Looking at the scalar field solution with these initial conditions, we define the maximum $r$ for which the solution is still exponentially decaying as $r_\infty$, where for $r > r_\infty$ the exponentially growing mode takes over. We set up a shooting method routine similar to the methodology described in Sec. 3.2 and Appendix C to find the initial conditions that match the solution to the asymptotic behavior (37) at $r_\infty$. We justify matching the solutions to the asymptotic limit for some finite $r_\infty \neq \infty$ by similar arguments as for the higher order perturbative solutions. In brief, $r_\infty$ is the maximum distance where the scalar field has essentially fallen off to 0. For a vanishing scalar field the metric is the Schwarzschild solution as described in Sec. 4.1.1, hence we can already require the metric function $A(r)$ and scalar field to follow (37) at $r_\infty$. Additionally as the constants $A'_\infty$, $\varphi'_\infty$ are unknown, we define our shooting method in terms of the ratios $e^A/(e^A)'$ and $\varphi/\varphi'(r)$ as functions of the initial conditions to

489   match

$$\frac{e^{A(r)}}{e^{A(r)'}} \to -r \ , \qquad \frac{\varphi(r)}{\varphi'(r)} \to -\frac{r}{(1+mr)} \ , \tag{43}$$

490   and determine $A'_\infty$, $\varphi'_\infty$ afterwards. We achieve this by defining a function of the differ-
491   ence between the metric solution with the initial conditions found as described above and the
492   asymptotic limit in (37), and similarly for the scalar field solution, as a function of $A'_\infty$ and
493   $\varphi'_\infty$ respectively.
494   To match the coefficients, we integrate over the absolute difference between the solution and
495   the asymptotic limit and determine the constants $A'_\infty$ and $\varphi'_\infty$ that minimize the integral over
496   a small region in $r$. For $A'_\infty$ the small region was determined around $r_\infty$ and for $\varphi'_\infty$ the re-
497   gion is based on integer multiples of the Compton wavelength. For each choice of parameters,
498   we require that the minimized integral is $\lesssim 10^{-9}$ as criterion for a good match, where $A'_\infty$ is
499   approximately constant and thus less sensitive to the choice of integral range than $\varphi'_\infty$, which
500   requires matching two functions that are rapidly decaying.
501   Additionally, we are interested in the solution for the spacetime inside the horizon to see if
502   the scalar field extends inside the horizon and to analyze the singular behavior of the space-
503   time inside the black hole. We therefore use an extension of the metric (11) as done in [47]
504   by defining a coordinate patch inside the horizon described by similar metric potentials as
505   in (11) but the opposite signs. With this convention, we capture the sign flip that occurs for
506   the time and radial components of the metric in Schwarzschild coordinates inside the horizon,
507   for which the time coordinate becomes spacelike and vice versa. This switch is then incorpo-
508   rated in the additional minus sign and therefore the solution to the metric corrections itself
509   can retain the same sign in- and outside the horizon. With this setup, we calculate numerical
510   solutions to (36) by integrating from a small distance inside the event horizon to $r = 0$. An
511   important assumption in this process needed to set the initial value of the scalar field is that
512   the limit of the scalar field approaching the horizon from both sides exists and can be glued
513   together smoothly. This implies that the same initial conditions and coefficients $\varphi_h$, $A'_h$ apply
514   as for the outside solution. However, the metric functions are discontinuous in this setup, for
515   instance, the solution for $A(r)$ diverges to minus infinity on both sides of the horizon.
516   A caveat is that the solution inside the horizon in Schwarzschild coordinates is not very mean-
517   ingful, for example, there is no intuitive interpretation of the coordinates. However we can
518   nevertheless use this solution to show that the scalar field extends to the inside of the black
519   hole and to analyze the behavior of curvature invariants inside the horizon. We compute and
520   discuss these curvature scalars in Sec. 6. As these quantities contain coordinate independent
521   information, the conclusions of our analysis are valid more generally beyond the particular
522   choice of interior coordinates.
523   In this way, we construct the full numerical solution for $A(r)$ and $\varphi(r)$ in and outside the hori-
524   zon. We compare this to the solution for a massless scalar field and the perturbative solution.

## 5   Comparison between massless, massive, and perturbative solutions

527   In Fig. 3 we show different results for the metric function $A(r)$ defined in (11) and in Fig. 4
528   the corresponding scalar field profile $\varphi(r)$ for a coupling function of $f(\varphi) = e^{2\varphi}/4$. The pink
529   curves correspond to the full solution with vanishing scalar field mass, while black curves are
530   the results for a mass of $\hat{m} = 0.1$. The upper panels are for a larger value of the coupling than
531   the lower ones. For the perturbative and Schwarzschild solutions we only show the curves
532   outside the black hole horizon.
533       Before discussing the results, we note an important point regarding comparisons between

the perturbative and exact solutions. The perturbative solutions are computed in terms of $u = r_S/r$ and similarly for the Schwarzschild solution. These need to be rescaled to compare with the full solution shown here in terms of $r_h/r$. We choose to compare black holes with the same ADM mass[4], which implies for the asymptotic limit of the full sGB solutions (37) that $A_\infty^{(1)}/r = r_S/r$. Next, we rescale the radial coordinate of the full solution such that $r_h = 1$. The ratio between the Schwarzschild and sGB horizons can be obtained via $r_S/r_h = A_\infty^{(1)}/1$ and is used to rescale the perturbative and Schwarzschild solution in the figures below.

### 5.0.1 Massless case: code check and singularity

First, we focus on the massless case as it has been more comprehensively studied in previous literature. For an independent check of our results, we compare the pink curves in Fig. 3 with a corresponding result in Fig. 1 of [47] and verify a similar qualitative behavior, up to small differences arising from different choices of coupling functions and -constant. Next, we analyze the features of the metric potential in Fig. 3 and corresponding scalar profile in Fig. 4. At large distances, they show the expected asymptotic behavior $A(r) \to 0$ and an exponential decay for the scalar field. Near the horizon (black vertical line), the scalar field remains finite while $A(r)$ diverges to minus infinity when approaching from the outside. For the coupling $\hat{\alpha} \sim 0.2$ shown in the upper panel of Fig. 3 the divergence for $r < r_h$ occurs very close to the horizon $r/r_h \sim 0.99$ (pink dashed line). This is due to the presence of a finite radius singularity, which is a well known phenomenon for massless sGB black holes [47, 48, 55]. We will make a concrete identification between this divergence in $A(r)$ and a genuine curvature singularity in Sec. 6.1.1. We see from the lower panel of Fig. 3 that for a smaller coupling $\hat{\alpha}$, the singularity moves further to the interior, as expected based on recovering the GR limit for zero coupling. This implies that the maximum value of $\hat{\alpha}$ for which a black hole exists is determined by the singularity coinciding with the horizon; higher values of $\hat{\alpha}$ will lead to a naked singularity.

### 5.0.2 Effect of the scalar mass

Qualitatively, the features of the solutions for finite scalar field mass are similar to the massless case. For the metric functions outside the horizon, the mass has a very small effect, as seen in Fig. 3, while for the scalar field in Fig. 4 the differences are more noticeable. The singularity for the massive case occurs at $r/r_h = 0.97$ for a coupling of $\hat{\alpha} = 0.2$. A larger mass of the scalar field thus shifts the singularity further inwards, as also expected from the infinite mass limit, where the scalar condensate disappears and the black hole reduces to Schwarzschild with a singularity at $r/r_h = 0$. This implies that the maximum value of the coupling for which black hole solutions exist increases for larger scalar field masses, consistent with the results of [71].

From a computational perspective we directly identify the maximum $\hat{\alpha}$ for black hole solutions based on the fact that for any value exceeding it, the near-horizon initial conditions and the asymptotically flat limit can no longer be connected by a smooth numerical solution.

### 5.0.3 Performance of the perturbative small-coupling solutions and comparison to Schwarzschild

Another interesting feature illustrated in Figs. 3 and 4 is the quality of the perturbative solutions to $O(\hat{\alpha}^3)$ corresponding to the green curves. We see that near the horizon for the larger value of the coupling (upper panel) the perturbative solution differs appreciably from the full

---

[4]For sGB and Schwarzschild black holes with the same ADM mass, the global mass generally differs due to the contributions from the scalar field in sGB [46]

solution. This is most noticeable when comparing the locations of the horizon, which for the perturbative and Schwarzschild solution lie at larger radial coordinate than the full solution, as indicated by the divergence of $A$ to $-\infty$. While the near-horizon behavior of the perturbative solution is based on expanding around the Schwarzschild horizon (29), the actual black hole horizon in this case is determined by the root of $\bar{B}$. After appropriately rescaling coordinates as described in the beginning of the section, this leads to the horizon locations indicated in the plots. As expected, for larger couplings the differences between the perturbative and exact solutions become larger, which is especially noticeable near the horizon. As mentioned, for larger couplings the singularity lies close to the horizon, and it is reasonable to expect non-perturbative effects to be important in its vicinity. In the large $r/r_h$ limit, the perturbative and numerical solutions coincide as the curvature effects become less and less significant. We also see that for the smaller coupling shown in the lower panel, the perturbative solution agrees much better with the full solutions near the horizon, as it is also farther from the singularity and the horizon moves closer to $r_h$. In Appendix D we give some additional analysis on the perturbative solution comparing also the solutions up to different orders in the coupling. Together with this analysis we conclude that the perturbative solution also becomes more accurate in the large scalar mass regime. As expected as in the large mass limit the singularity shifts inwards further away from the horizon. Furthermore, we find no particular behavioural change comparing the solution up to quadratic and cubic order, for which the metric corrections to scalar field come in, see Table 1. Lastly we find the difference of the perturbative solution in the near horizon region to be largest. However even with the finite radius singularity lying close to the horizon for larger values of the coupling, when restricting to the regimes away from the immediate vicinity of the divergence, we find no sign of qualitatively new non-perturbative behaviour that would not be approximately captured by adding higher small coupling corrections to the solution.

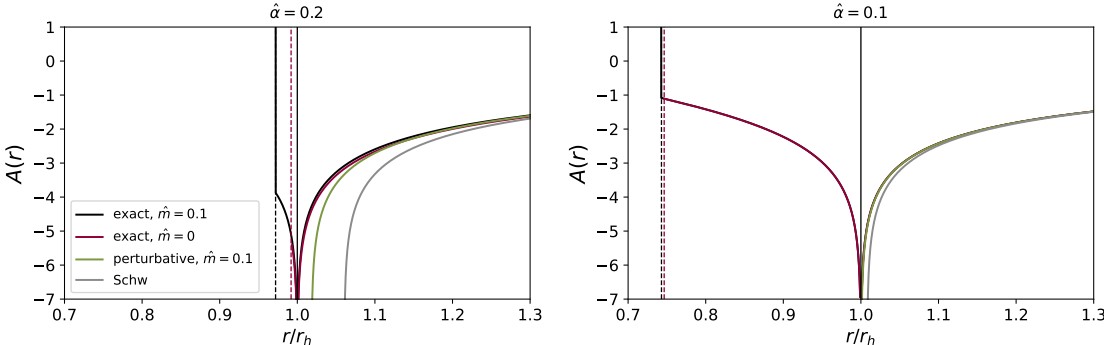

Figure 3: *Behavior of the metric function $A(r)$ characterizing the time-time component of the metric* for couplings of $\hat{\alpha} = 0.2$ (upper panel) and $\hat{\alpha} = 0.1$ (lower panel). Black curves show the full solution for a scalar field mass $\hat{m} = 0.1$, pink curves the massless case, green curves represent the perturbative solution including corrections to $O(\hat{\alpha}^3)$ and grey curves show the Schwarzschild solution for comparison. For the latter two only the solutions outside the horizon are shown. The black vertical line denotes the horizon radius and the vertical dashed curves the singularities.

## 6 Properties of the solutions

Having constructed the full and perturbative numerical solutions for a static black hole in massive sGB, we analyse the properties of these solutions. We start by studying the spacetime

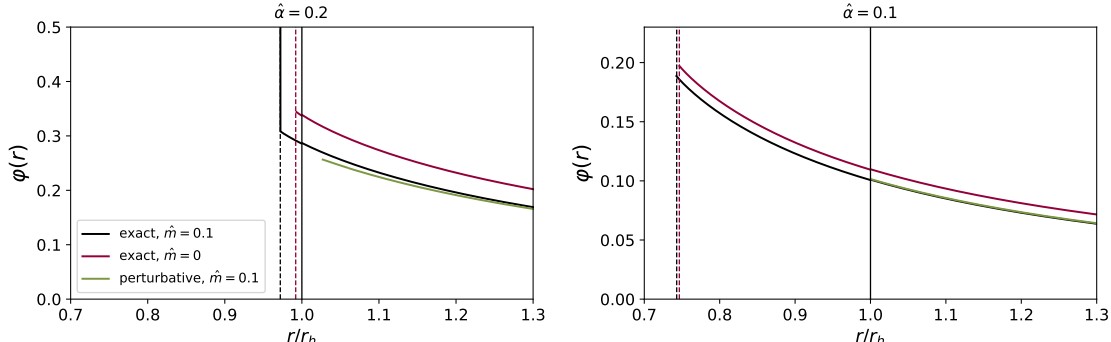

Figure 4: *Scalar field profile* for couplings of $\hat{\alpha} = 0.2$ (upper panel) and $\hat{\alpha} = 0.1$ (lower panel). Black curves show the full solution for a scalar field mass $\hat{m} = 0.1$, pink curves are for the massless case, and green curves represent the perturbative solution including corrections to $O(\hat{\alpha}^3)$ only showing the solution outside the horizon. The black vertical line denotes the horizon radius and the vertical dashed curves the singularities.

curvature in and outside the horizon and recover how properties such as the amount of scalar field on the horizon or the scalar monopole charge depend on the parameters of the theory. The analysis in this section complements the discussion of [71] which focused on the horizon radius, amount of scalar field at the horizon, black hole surface, entropy, and temperature as function of the black hole mass for different coupling functions and scalar field masses. Note that the rescalings in [71] to obtain dimensionless variables are different from those used in this paper, in particular, we rescale based on the horizon radius, while [71] rescaled by the coupling constant. In all further analysis we specify to a dilatonic coupling function $f(\varphi) = e^{2\varphi}/4$.

## 6.1   Characterizing the curvature and field density

Before we analyze more specifically how certain properties of the black hole solutions depend on the parameters of the theory, we first consider the curvature scalars and energy density around the black hole to gain more intuition for the solutions.

### 6.1.1   Curvature invariants and singularity

To characterize the curvature we analyze the curvature invariants. Here we focus on the Kretschmann scalar

$$\mathcal{K} = R_{\mu\nu\rho\sigma}R^{\mu\nu\rho\sigma} \ , \tag{44}$$

and its cousin; the fully contracted Weyl tensor squared

$$\mathcal{C} = C_{\mu\nu\rho\sigma}C^{\mu\nu\rho\sigma} \ . \tag{45}$$

In vacuum in GR these two invariants coincide. We calculate them using the full numerical solution for a coupling of $\hat{\alpha} = 0.2$ and for masses of $\hat{m} = 0.1$ and $\hat{m} = 1$. The results are illustrated in Fig. 5, where the bottom panel shows the percent difference of the Kretschmann scalar for a massive sGB and Schwarzschild black hole. Here we make the same choice as for Fig. 3, comparing to a Schwarzschild black hole with the same ADM mass. We see that the curvature invariants blow up for $r/r_h \sim 0.88$ and $r/r_h \sim 0.97$ for $\hat{m} = 1$ and $\hat{m} = 0.1$ respectively. For $\hat{m} = 0.1$ this corresponds to the same location as the divergences in $A$ and

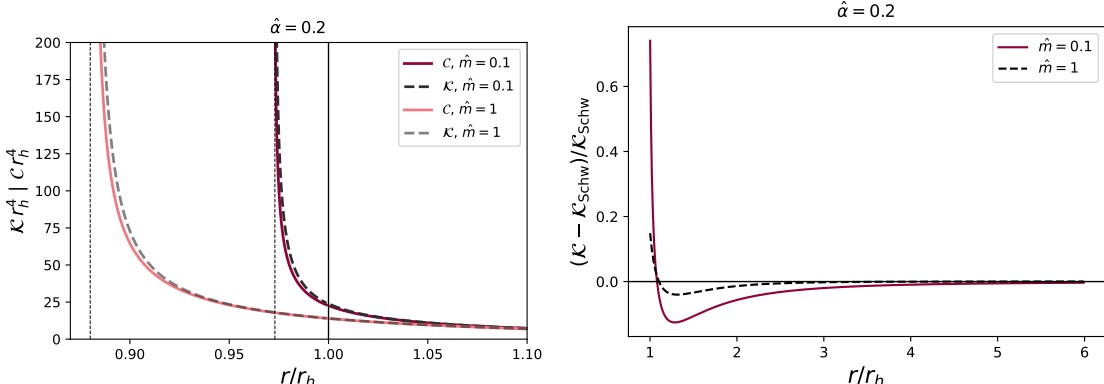

Figure 5: Top panel: *Curvature invariants* of the Kretschmann scalar $\mathcal{K}$ (dashed lines) and the contracted Weyl tensor $\mathcal{C}$ (pink and purple curves) in- and outside the event horizon (black vertical line) for two values of the scalar field mass (lighter colors for larger mass). The vertical dashed black lines denote the location of the finite radius singularity. Bottom panel: The *percent difference* of $\mathcal{K}$ in massive sGB and Schwarzschild for two values of the scalar field mass only for the spacetime outside the horizon.

$\varphi$ seen in the top panels in Fig. 3 and 4, which corroborates the identifications between these divergences and genuine singularities already mentioned in Sec. 4. We also note from comparing the solid curves corresponding to $\mathcal{C}$ and the dashed lines illustrating the results for $\mathcal{K}$ in Fig. 5 that while for most regions outside the horizon the two kinds of curvature invariants coincide, they differ slightly in its immediate vicinity and the interior.

Looking at the bottom panel of Fig. 5 we see that close to the horizon up to $r/r_h \sim 1.1$, the curvature in sGB gravity is larger than for the Schwarzschild black hole. Interestingly, however, in the region $1.1 \lesssim r/r_h \lesssim 5$ the curvature in sGB is weaker than Schwarzschild, with the fractional difference attaining its largest negative value around $r/r_h \sim 1.3$. In the large-$r$ limit the curvature invariants coincide, as expected. With increasing scalar field mass, the curvature decreases. Hence, the massless limit leads to the strongest curvature and thus largest deviation from Schwarzschild. The distinguishability of the curvature up to $r/r_h \sim 5$ could have interesting consequences, for instance, for tidal effects.

### 6.1.2 Energy density

The energy density of the spacetime is given by $T_t^t = -\rho$ in (47). Additionally we define $\rho_\varphi$ as the pure scalar contributions of $T_t^t$ which can be obtained by setting $\alpha \to 0$ in (47). The results for the energy densities for a case with the maximum coupling for a massless sGB black hole are illustrated in Fig. 6. The top panel shows the full energy density including the higher curvature contributions for a scalar mass $\hat{m} = 0.8$, while the bottom panels show the corresponding radial profiles for that case (green curves) and the massless one (black curves).

The upper panel of Fig. 6 shows that $\rho$ is concentrated close to the horizon and becomes more dilute further away from the black hole. Around $r/r_h \sim 2$ the energy density has already fallen off to essentially zero. From the bottom panels of Fig. 6, we see that for the full energy density (top), the same behavior occurs in the massless case, also around the same values. However, the pure scalar field contribution to the energy density (bottom) has very different features, namely for the massless field configuration, the falloff to zero is much slower, as expected based on the asymptotic behavior of the field (37) indicating the scalar field is suppressed for distances larger than the Compton wavelength. Specifically, the percent difference

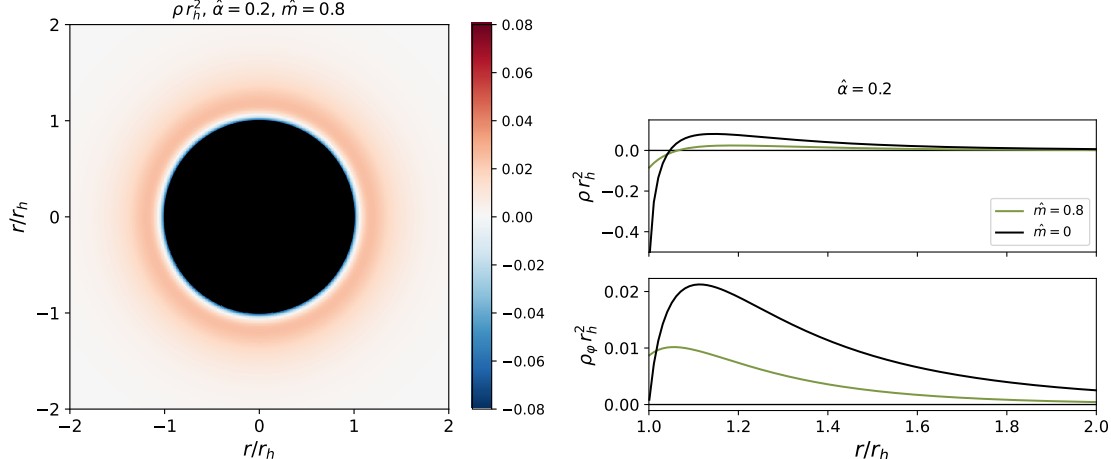

Figure 6: Top panel: *2D figure of the energy density* around the black hole shown as the black disk for a system with $\hat{\alpha} = 0.2$, $\hat{m} = 0.8$. Bottom panel: *Energy density (top) and scalar part of the energy density (bottom)* as function of $r$ for a massless and $\hat{m} = 0.8$ scalar field.

in $\rho_\varphi$ between one and two times the Compton wavelength ($\lambda_\varphi = 1.25$) for the massive case is **99%**. For comparison, in the massless case, the falloff of the density between the same radial distances is only **94%**. Another interesting feature is that while the scalar field contribution is always positive, the full energy density is not. The reason is that the higher curvature contributions can have different signs, which leads to a negative total energy density close to the black hole horizon. The fact that the energy density can become negative is one of the reasons black holes both in massless and massive sGB can evade the no hair theorem [25, 28].

## 6.2 Scalar hair, regularity constraint and bound on the coupling

As explained in Sec. 4.1, requiring the scalar field solution to be regular at the horizon leads to a constraint for the derivative of the scalar field at the horizon, c.f. (24), (26) and (40) for the linear-in-coupling, massless and massive full theory respectively. From Fig. 4 we conclude that for the scalar field solution near the horizon to be able to match the asymptotic fall off, the derivative at the horizon needs to be real and negative in terms of $r$ or positive for $u$. For the linearized case this is accomplished via (25) and in the massless full theory case this is done by imposing the square root to be real via (27). However, in the massive case requiring the square root to be real by imposing $C > 0$ does not ensure $\varphi'_h < 0$ (40). Therefore in this case both $C > 0$ and $\varphi'_h < 0$ need to be imposed to ensure an asymptotically flat solution. All of these inequalities depend on the parameters $\varphi_h$, $\hat{\alpha}$ and $\hat{m}$. The dependence on $r_h$ is encapsulated in the dimensionless parameters $\hat{\alpha}$, $\hat{m}$. In this section, we study these inequalities imposed near the horizon to determine how $\varphi_h$ depends on the theory parameters.

The top panel of Fig. 7 compares the results of the near-horizon constraints on $\varphi_h$, indicated by the solid (linearized) and dashed (full) lines to the values extracted from the numerical solution in the linearized (diamonds) and massive full theory (dots) cases. We see that the amount of scalar hair at the horizon and the difference between the linearized and full theory results is largest for a larger coupling, as expected. For scalar field masses larger than $\hat{m} > 1$, where in dimensionfull parameters the Compton wavelength lies inside the black hole horizon, the scalar hair is severely suppressed (note the logarithmic scale of the plot). In the large $\hat{m}$ limit, the linearized and full theory result coincide as for $\hat{m} \to \infty$ the scalar field should decouple and black holes should have no hair.

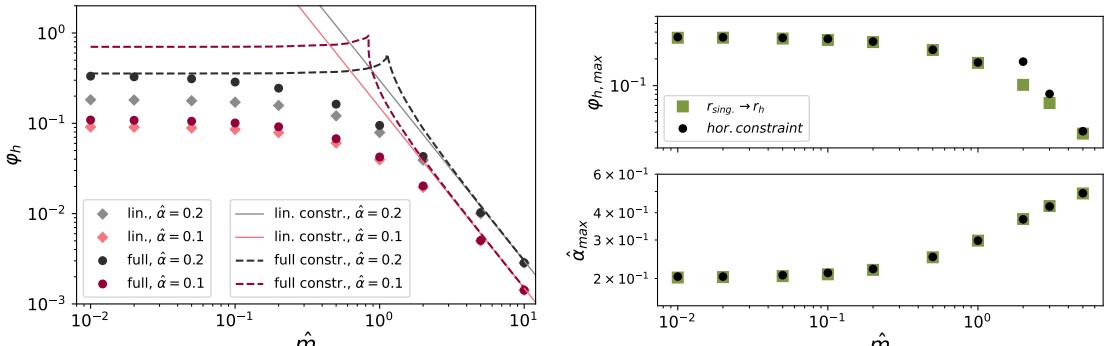

Figure 7: Top panel: The amount of *scalar field at the horizon* $\varphi_h$ as function of the dimensionless scalar field mass $\hat{m}$. The panel shows the values for the linearized (diamonds) and full solution (dots) of the scalar field equation for two different values of the coupling (grey and pink shades). Additionally, the linearized (25) (solid lines) and full theory constraints (40) (dashed lines) are shown for two different values of the coupling. Bottom panels: *maximum allowed value of the scalar field at the horizon and coupling constant* as a function of the scalar field mass from the requirement of preventing a naked singularity (squares) and the near horizon constraint (40) (dots).

In the same panel we also show the linearized inequality (25) as the pink and gray lines. We obtain the full theory constraint (dashed lines) by selecting the largest value for $\varphi_h$ allowed for which (40) is real and negative for each choice of $\hat{\alpha}$ and $\hat{m}$. For increasing $\hat{m}$, we find that beyond a certain coupling-dependent threshold that coincides with the cusp in the dashed curves in Fig. 7, two branches of values for $\varphi_h$ arise for which $\varphi'_h < 0$. For one branch the values of $\varphi_h$ becomes larger for larger mass while for the other branch they become smaller, which we identify as the desired physical solution. We therefore selected the largest possible $\varphi_h$ in the physical branch. From Fig. 7 we see that the linearized and full constraints coincide in the large mass limit as required. The values for $\varphi_h$ obtained from the numerical solutions are always below the curves from the near-horizon constraints. In the small-mass limit, the matching to the asymptotic falloff fixes $\varphi_h$ to smaller values than allowed by the near-horizon constraints. In the zero-mass limit and largest possible coupling in the massless theory $\hat{\alpha} \sim 0.2$, the amount of scalar field on the horizon approaches the largest allowed value by the near-horizon constraint. Similarly, in the large-mass limit, the numerical solution for $\varphi_h$ approaches the maximum allowed value by the corresponding near-horizon constraint.

The literature on the massless theory suggests that the near-horizon constraint (26) prevents the finite surface singularity from extending outside the black hole horizon. We analyze the link between the singularity and the near-horizon constraint in the massive theory in the bottom panels of Fig. 7. These plots show the results of the following procedure. For a fixed mass $\hat{m}$, we increased $\hat{\alpha}$ up to the value for which the curvature singularity lies on the horizon. This identifies the largest possible $\hat{\alpha} = \hat{\alpha}_{\text{max}}$ to prevent a naked singularity and a corresponding $\varphi_h^{\hat{\alpha}_{\text{max}}}$. For the same $(\hat{\alpha}_{\text{max}}, \hat{m})$, we also determined the maximum allowed $\varphi_h$ for which $\varphi'_h < 0$ from (40). These two results for the maximum allowed $\varphi_h$ are shown as the green squares (singularity constraint) and black dots (horizon constraint) in the middle panel of Fig. 7. Next, we considered the implications of the horizon constraint (40) when evaluated for the maximum hair $\varphi_h^{\hat{\alpha}_{\text{max}}}$ set by the verge of a naked singularity to determine the corresponding maximum allowed coupling $\hat{\alpha}$ for which $\varphi'_h < 0$. These results for the maximum coupling, together with those obtained from the singularity constraint are shown in the bottom panel of Fig. 7 as black dots and green squares respectively.

We see from Fig. 7 that for masses $\hat{m} < 1$, the constraints from the curvature singularity and regularity of the field at the horizon on the maximum $\varphi_h$ coincide. For slightly larger masses, the cusp feature arises in the horizon constraint, as discussed above. In this regime, the requirement of not having a naked singularity is a stronger constraint than the near horizon requirement. For the maximum allowed coupling, both cases agree for the range of masses we studied, hence the maximum coupling is not sensitive to the choice of $\varphi_h$.

### 6.2.1 Implications in relation to the coupling and scalar field mass

The results from the bottom panel on Fig. 7 show the theoretical bound on the coupling as function of the scalar field mass. As mentioned in Sec. 2.1 a first observational constraint on the coupling is $\sqrt{\alpha} \lesssim 2.47$km for $10^{-15}$eV $\lesssim m \lesssim 10^{-13}$ [78]. Relating this to the dimensionless mass defined in (17) for stellar mass black holes ranging from approximately $5M_\odot \lesssim M \lesssim 150M_\odot$ with horizon size of order $r_h \sim r_S$, this observational constraint on the coupling is set for $10^{-5} \lesssim \hat{m} \lesssim 10^{-1}$. Notably, this implies that for stellar mass black holes, the tightest constraints on the coupling in the range of masses $\hat{m} > 10^{-1}$ are given by the theoretical constraints shown in the bottom panel of Fig. 7.

We can also use the results of the top panel of Fig. 7 to make a rough estimate of the possible scalar field mass range that would be interesting in relation to observation. From Fig. 7 we find that beyond $\hat{m} \sim 1$ the scalar field becomes highly suppressed, which decreases the likelihood for detection by probing the black hole environment. Hence $\hat{m} \sim 1$ seems the largest scalar field mass for which there is still significant scalar hair around the black hole. Then we consider a back of the envelope calculation similar to what was done in [88]. We assume that astrophysical black holes lie in the mass range $5M_\odot \lesssim M \lesssim 10^{10}M_\odot$ again using $r_h \sim r_S$ to find the dimensionful mass via (17), (4) for $\hat{m} = 1$. This results in

$$1.3 \times 10^{-11}\text{eV} \gtrsim m_\varphi \gtrsim 6.7 \times 10^{-21}\text{eV} \,. \tag{46}$$

Massive sGB black holes thus enable exploring a large swath of parameter space of ultralight dark matter models, see e.g. [60] for a review.

## 6.3 Dependencies of black hole properties

### 6.3.1 Innermost stable circular orbit and light ring

Next, we use the full numerical solutions to analyze the dependence of gauge-invariant quantities such as the orbital frequency of a test particle at the innermost stable circular orbit (ISCO) and a photon at the unstable circular orbit (light ring) on the parameters of the theory.

In Appendix E we compute the ISCO and light ring (LR) radii from considering geodesic motions of test particles and photons, and formulating the dynamics in terms of an effective potential whose maximum determines the ISCO and LR. Specifically, we calculated the roots of the second derivative of (59) and (64) numerically after substituting the solutions for $A(r)$ and $B(r)$. We convert all expressions to functions of the orbital frequency as it is a coordinate-independent quantity by contrast to the radius, by using the relationship between the radial coordinate and frequency from (61). In Fig. 8 we show the difference between the orbital frequency $\omega$ at the ISCO/LR in massive sGB and Schwarzschild spacetimes for different scalar field masses. Note that we give the results in terms of the dimensionless quantity $\omega r_h$, therefore the Schwarzschild frequencies $\omega_{ISCO} r_S = 1/3\sqrt{6}$, $\omega_{LR} r_S = 2/3\sqrt{3}$, need to be rescaled to $r_h$ in the same way as described in Sec. 5.0.3.

From both panels of Fig. 8 we conclude that, as the differences are positive, the orbital frequencies in massive sGB are larger (corresponding to the ISCO/LR radii being smaller) than for a Schwarzschild black hole with the same ADM mass. When comparing this to our

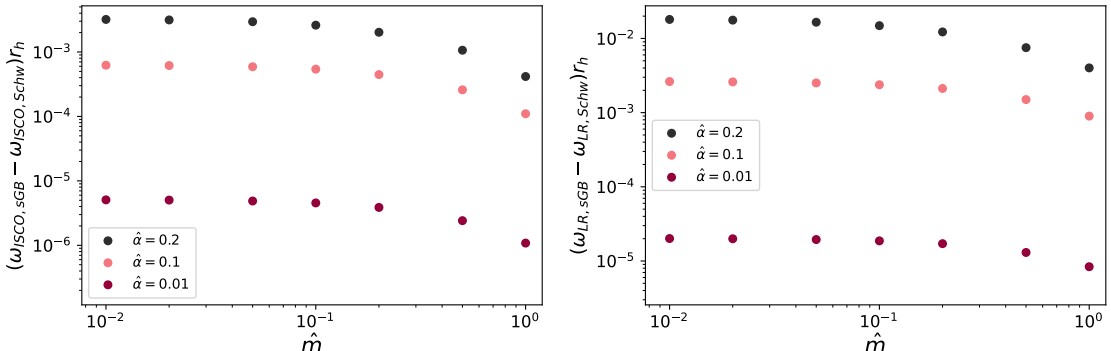

Figure 8: *The difference in orbital angular frequency at the ISCO (top panel) and LR (bottom panel) rescaled by the horizon radius from the Schwarzschild results as a function of the scalar mass for three different values of the coupling.*

result for the behavior of the curvature in the bottom panel of Fig. 5, we see that for radii around $r_{ISCO}/r_h \sim 3$, $r_{LR}/r_h \sim 3/2$ the curvature in massive sGB is less strong than for a Schwarzschild black hole, and a stable orbit for a test particle/photon can therefore lie closer to the horizon. This also corresponds to the findings for the ISCO/LR frequencies in the literature on massless sGB, e.g. in [89, 90]. Furthermore, Fig. 8 shows that the difference in the orbital frequencies becomes smaller for smaller coupling and larger masses, as expected in these limits. As for other quantities, massless sGB gives the strongest deviations from a Schwarzschild blackhole.

### 6.3.2   ADM mass and scalar charge

Lastly we consider the analysis of the obtained ADM mass $M_{\text{ADM}} = 1/2 A'_\infty$ and scalar monopole charge $\varphi'_\infty$ defined in (37). We obtain these quantities from the numerical solutions as described in Sec. 4.2 for different masses and coupling. The ADM mass and scalar charge are relevant e.g. in effective action descriptions for black hole binary systems [55], where the two bodies are reduced to center-of-mass worldlines augmented with additional parameters that are matched to physical properties of the full configuration and capture its coarse-grained effects. In the massless case, the scalar charge is defined to be the coefficient of the $1/r$ term in the asymptotic falloff of the scalar profile. However, for massive scalar fields the asymptotic limit has an exponential decay (37) and the definition of the charge must be adapted. We consider here the convention of [65], which is still based on the decaying tail of the scalar field solution and defines the charge to be the prefactor of the exponential $\varphi'_\infty$ as given in (37). As described in Sec. 4.2, matching the solution to the asymptotic limit for the scalar field is more susceptible to the choice of integral region used for the matching than the metric functions. In practice we therefore limited the construction of $\varphi'_\infty$ to $\hat{m} \le 1$, as for larger values the solution outside the black hole horizon has already fallen off to nearly zero and it is not possible to unambiguously match to (37) to determine $\varphi'_\infty$. In principle one could obtain the charge in this regime by working with the solution in the interior of the horizon, however as we mentioned in Sec. 6.2 the $\hat{m} \lesssim 1$ regime is the most interesting, therefore we limited our analysis to this regime. We show the results of these calculations for the ADM mass and scalar charge as function of the scalar field mass in Fig. 9. From the Fig. 9 we see that both the scalar charge and ADM mass become less sensitive to the scalar field mass for smaller values of the coupling. Both are also proportional to the coupling, where for vanishing coupling constant, the scalar charge vanishes and the ADM mass goes to $1/2 r_h$ as expected.

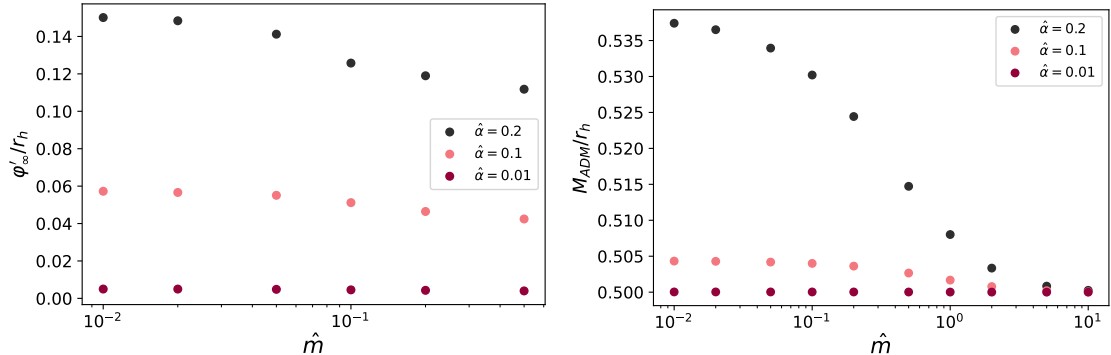

Figure 9: *The amount of scalar charge (top panel) and ADM mass (bottom panel) as function of the dimensionless scalar field mass $\hat{m}$ for three different values of the coupling.*

## 7 Conclusion

In this paper we performed a systematic study of various features of static, spherically symmetric black holes in sGB with a massive scalar field. This is a more natural scenario than assuming a massless field, as has been the focus of the majority of previous work, except for a few numerical examples. The scalar field mass introduces an additional scale in the problem and gives rise to richer features of the spacetime and scalar condensate. For the first time, we calculated perturbative solutions in a small coupling expansion up to third order in $\hat{\alpha}$ and compared this to full numerical solutions for the spacetime and scalar field. The small-coupling approximation yields more direct analytical insights into intriguing features that arise, while the numerical solutions capture fully nonlinear regimes. To compute numerical solutions, we used a bisection method to approach the scalar field solution with the desired asymptotic fall-off behavior at spatial infinity and a shooting method to obtain the metric potentials with the correct near-horizon and asymptotic behaviors. By extending the full numerical solutions inside the horizon, we found that the metric potentials and scalar field diverge at a finite radius. From analyzing the Kretschmann and contracted Weyl tensor curvature invariants we concluded that these divergences coincide with a genuine curvature singularity. The location of this singularity depends on the coupling constant and the scalar field mass, where for smaller couplings and higher masses the singularity moves closer to the center of the black hole. The location of the singularity also impacts the performance of the small-coupling perturbative solution, which we found to be viable for small couplings, large scalar masses, and large distances from the black hole. These trends can be attributed to the fact that at large distances and for small coupling, the scalar and nonlinear curvature effects decrease, and for large scalar masses the scalar field decouples from the metric and the GB contribution to the action becomes a total derivative with no dynamical impact.

For finding the black hole solution, in addition to the condition of asymptotic flatness one requires the scalar field to be regular at the horizon. This leads to conditions relating the amount of scalar field at the horizon to the coupling constant, black hole radius and scalar field mass. We discussed these conditions in the massless, linearized, and exact cases and interpreted them for the maximum amount of scalar hair possible at the horizon for each choice of parameters. Inverting this argument led to an upper bound on the coupling or lower bound on the mass of the black hole. We found that introducing the scalar field mass leads to the existence of black hole solutions for larger couplings or conversely, for a fixed coupling, the domain of black holes extends to lower masses, see also [71]. By comparing these near-horizon conditions for the scalar field to the parameters corresponding to the finite radius singularity being

located at the black hole horizon, we found that both lead to bounds on the maximum scalar field at the horizon. The latter condition provides a more stringent constraint on the maximal possible $\varphi_h$ when the Compton wavelength is comparable to the radius of the black hole. For the maximum allowed value of the coupling constant, the near horizon condition and the singularity bound agree well. Comparing these theoretical bounds on the coupling with the first observational constraint on massive sGB [78] we concluded that for stellar mass black holes, for masses $\hat{m} > 10^{-1}$ our theoretical results provide the most stringent bounds to date. Additionally, in the case of the scalar Compton wavelength being larger than the black hole radius, the numerical matching between the near horizon and asymptotic limits shows that the actual values lie well below these bounds. On the other hand, for Compton wavelengths smaller than the black hole radius, the amount of scalar hair at the horizon in the exact calculations turns out to be identical to the maximum value determined by the regularity constraint.

Using the results for the amount of scalar hair near the horizon we could make a rough estimation of the scalar field mass range that could be promising in the light of observation. We found that this mass range includes the current scalar particle models. Lastly we analysed how the ISCO radius, light ring radius, ADM mass and scalar monopole charge depend on the scalar field mass and coupling constant.

For future work, the study of black holes in sGB could be extended to include next to the scalar field mass, also the self interaction term in the scalar potential or one could add different interactions e.g. study the optical channel, including the interaction with light. Considering massive scalar fields in other promising modified gravity contexts would be interesting as well. Furthermore, obtaining rotating black hole solutions would be a next exploration. This would lead to the opportunity to study the possibility of superradiance instability in the context where the massive scalar field is also coupled to the spacetime curvature. Our work contributed to the first exploratory studies of sGB black holes with massive scalar fields. The full and perturbative numerical solutions can be used in further studies of black holes in massive sGB and in modelling compact binary systems in these theories, extending the work of [79–81] to include massive scalar fields. Furthermore our analysis and numerical method related to the massive scalar cloud configuration can be applied to massive scalar fields in a broader context and our results on the ADM mass and scalar monopole charge can be useful in the effective field theory description of compact binaries. In direct continuation of this work the analysis of the gravitational radiation from compact objects in massive sGB can be explored, contributing to the efforts of probing the strong field environments of black holes in the search for beyond GR signatures.

*Acknowledgments.* We thank Gastón Creci, Peter Jonker and Ramon Wakelkamp for useful discussions. This publication is part of the Dutch Black Hole Consortium with project number NWA.1292.19.202 of the research programme NWA which is (partly) financed by the Dutch Research Council (NWO).

## A  Explicit expressions for equations

In this appendix we show the explicit equations that were not given in the main text for the sake of readability.

### A.0.1  Field equations

The explicit components of the modified Einstein equations (8) with (9) in terms of the metric (11) are given by

$$
\begin{aligned}
G^t_t =& -\frac{1}{r^2} + \frac{e^{-B(r)}}{r^2} - \frac{e^{-B(r)}B'(r)}{r}, \\
G^r_r =& -\frac{1}{r^2} + \frac{e^{-B(r)}}{r^2} + \frac{e^{-B(r)}B'(r)}{r}, \\
G^\theta_\theta =& G^\varphi_\varphi = \frac{e^{-B(r)}\Big(rA'(r)^2 - 2B'(r) + A'(r)(2 - rB'(r) + 2rA''(r))\Big)}{4r}, \\
T^t_t =& -\frac{e^{-2B(r)}}{r^2}\Big[\varphi'^2\big(r^2 e^{B(r)} + 4\alpha f''(\varphi)(e^{B(r)} - 1)\big) - 2\alpha f'(\varphi)\big(B'(r)\varphi'(r)(e^{B(r)} - 3) \\
& -2\varphi''(r)(e^{B(r)} - 1)\big) + \big(e^{B(r)}m r \varphi(r)\big)^2\Big], \\
T^r_r =& \frac{e^{-B(r)}\varphi'(r)}{}\left[\varphi'(r) - \frac{2e^{-B(r)}\big(e^B(r) - 3\big)\alpha f'(\varphi)A'(r)}{r^2}\right] - (m\varphi(r))^2, \\
T^\theta_\theta =& T^\varphi_\varphi = -\frac{e^{-2B(r)}}{r^2}\Big[\varphi'^2(r)\big(r e^{B(r)} - 2\alpha f''(\varphi)A'(r)\big) - 4\alpha f'(\varphi)\big(A'^2(r)\varphi'(r) \\
& + 2\varphi'(r)A''(r) + A'(r)\big(2\varphi''(r) - 3B(r)'\varphi'(r)\big)\big) + e^{2B(r)}m^2 r^2 \varphi(r)^2\Big].
\end{aligned}
\tag{47}
$$

The scalar field equation (10) becomes

$$
\begin{aligned}
& 2r\varphi''(r) + \big(4 + rA'(r) - rB'(r)\big)\varphi'(r) + \frac{\alpha f'(\varphi)e^{-B(r)}}{r}\Big[\big(e^{B(r)} - 3\big)A'(r)B'(r) \\
& -\big(e^{B(r)} - 1\big)\big(2A''(r) + A'^2(r)\big)\Big] - 2e^{B(r)}m^2 r\varphi(r) = 0.
\end{aligned}
\tag{48}
$$

### A.0.2  Master equations in $A$ and $\varphi$

In Sec. 4, we rewrote the modified Einstein equations as a system of second order differential equations (36), where the right hand sides are given by the functions

$$f(r, \varphi(r), \varphi'(r), A'(r)) =$$

$$\Bigg( 4e^{4B(r)} m^4 \varphi(r)^3 \left( e^{B(r)} r - 4\alpha f'(\varphi) \varphi'(r) \right) r^4 - 8e^{4B(r)} \left( -1 + e^{B(r)} \right) m^4 \alpha \varphi(r)^4 f'(\varphi) r^3$$

$$- e^{2B(r)} m^2 \varphi(r) \left( e^{B(r)} r - 4\alpha f'(\varphi) \varphi'(r) \right) \left( 4e^{B(r)} \left( r^2 \varphi'(r)^2 + e^{B(r)} - 1 \right) - 3A'(r) \right.$$

$$\left. \left( e^{B(r)} r + 2 \left( -3 + e^{B(r)} \right) \alpha f'(\varphi) \varphi'(r) \right) \right) r^2 + 8e^{2B(r)} \varphi'(r) \left( e^{B(r)} r + \left( -5 + e^{B(r)} \right) \alpha f'(\varphi) \varphi'(r) \right)$$

$$\left( r^2 \varphi'(r)^2 + e^{B(r)} - 1 \right) r + 2e^{2B(r)} m^2 \varphi(r)^2 \left( -e^{2B(r)} \left( rA'(r) + 4 \right) \varphi'(r) r^3 - 6 \right.$$

$$\left( 3 - 4e^{B(r)} + e^{2B(r)} \right) \alpha^2 A'(r) f'(\varphi)^2 \varphi'(r) + \alpha f'(\varphi) \left( 4e^{B(r)} \left( \left( -1 + e^{B(r)} \right)^2 + 4r^2 \varphi'(r)^2 \right) \right.$$

$$\left. \left. + rA'(r) \left( 4\varphi'(r)^2 \left( e^{B(r)} r^2 + 2 \left( -1 + e^{B(r)} \right) \alpha f''(\varphi) \right) - 5e^{B(r)} \left( -1 + e^{B(r)} \right) \right) \right) \right) r$$

$$+ e^{B(r)} \alpha A'(r)^3 f'(\varphi) \left( r - 4\alpha f'(\varphi) \varphi'(r) \right) \left( e^{B(r)} r + 2 \left( -3 + e^{B(r)} \right) \alpha f'(\varphi) \varphi'(r) \right)$$

$$+ e^{B(r)} A'(r) \left( e^{B(r)} \varphi'(r) \left( \left( e^{B(r)} r^2 - 4\alpha f''(\varphi) \right) \varphi'(r)^2 + 2e^{B(r)} \left( -4 + e^{B(r)} \right) \right) r^3 \right.$$

$$- 12 \left( 15 - 8e^{B(r)} + e^{2B(r)} \right) \alpha^2 f'(\varphi)^2 \varphi'(r)^3 r + 4\alpha f'(\varphi) \left( -r^2 \left( e^{B(r)} r^2 + 4 \left( -2 + e^{B(r)} \right) \alpha f''(\varphi) \right) \varphi'(r)^4 \right.$$

$$\left. \left. - 2 \left( 3e^{B(r)} \left( -3 + e^{B(r)} \right) r^2 + \left( 1 - 3e^{B(r)} + 2e^{2B(r)} \right) \alpha f''(\varphi) \right) \varphi'(r)^2 + e^{B(r)} \left( -1 + e^{B(r)} \right)^2 \right) \right)$$

$$- A'(r)^2 \left( e^{3B(r)} \varphi'(r) r^4 + 2e^{B(r)} \alpha f'(\varphi) \left( \left( e^{B(r)} \left( -4 + e^{B(r)} \right) r^2 - 2 \left( -5 + 3e^{B(r)} \right) \alpha f''(\varphi) \right) \varphi'(r)^2 \right.$$

$$\left. + e^{B(r)} \left( -1 + e^{B(r)} \right) \right) r + 4\alpha^2 f'(\varphi)^2 \varphi'(r) \left( e^{B(r)} \left( 3 - 4e^{B(r)} + e^{2B(r)} \right) - 2\varphi'(r)^2 \right.$$

$$\left. \left. \left. \left( e^{B(r)} \left( -3 + 2e^{B(r)} \right) r^2 + \left( 9 - 8e^{B(r)} + 3e^{2B(r)} \right) \alpha f''(\varphi) \right) \right) \right) \right) \Bigg/ \Bigg( -4e^{2B(r)} \right.$$

$$\left( e^{B(r)} r - 4\alpha f'(\varphi) \varphi'(r) \right) \left( r^2 \varphi'(r)^2 + e^{B(r)} - 1 \right) r^2 + 4e^{2B(r)} m^2 \varphi(r)^2 \left( e^{B(r)} r^2 \right.$$

$$\left. \left( e^{B(r)} r - 4\alpha f'(\varphi) \varphi'(r) \right) - 4 \left( -1 + e^{B(r)} \right) \alpha^2 A'(r) f'(\varphi)^2 \right) r^2 - 8 \left( -1 + e^{B(r)} \right)$$

$$\alpha^2 A'(r)^2 f'(\varphi)^2 \left( e^{B(r)} r + \left( -9 + 5e^{B(r)} \right) \alpha f'(\varphi) \varphi'(r) \right) + e^{B(r)} A'(r) \left( 3e^{2B(r)} r^4 + 8e^{B(r)} \right.$$

$$\left. \left( -4 + e^{B(r)} \right) \alpha f'(\varphi) \varphi'(r) r^3 + 4\alpha^2 f'(\varphi)^2 \left( 5 \left( -1 + e^{B(r)} \right)^2 - 4 \left( -4 + e^{B(r)} \right) r^2 \varphi'(r)^2 \right) \right) \Bigg)$$

$$\tag{49}$$

874 and

$$
h(r, \varphi(r), \varphi'(r), A'(r)) =
$$

$$
\Big( 4e^{4B(r)} m^4 \varphi(r)^3 \left( e^{B(r)} r - 4\alpha f'(\varphi) \varphi'(r) \right) r^4 - 8e^{4B(r)} \left( -1 + e^{B(r)} \right) m^4 \alpha \varphi(r)^4 f'(\varphi) r^3
$$

$$
- e^{2B(r)} m^2 \varphi(r) \left( e^{B(r)} r - 4\alpha f'(\varphi) \varphi'(r) \right) \left( 4 e^{B(r)} \left( r^2 \varphi'(r)^2 + e^{B(r)} - 1 \right) - 3A'(r)
$$

$$
\left( e^{B(r)} r + 2 \left( -3 + e^{B(r)} \right) \alpha f'(\varphi) \varphi'(r) \right) \right) r^2 + 8 e^{2B(r)} \varphi'(r) \left( e^{B(r)} r + \left( -5 + e^{B(r)} \right) \alpha f'(\varphi) \varphi'(r) \right)
$$

$$
\left( r^2 \varphi'(r)^2 + e^{B(r)} - 1 \right) r + 2 e^{2B(r)} m^2 \varphi(r)^2 \left( -e^{2B(r)} \left( r A'(r) + 4 \right) \varphi'(r) r^3 - 6 \left( 3 - 4 e^{B(r)} \right.
$$

$$
+ e^{2B(r)} \right) \alpha^2 A'(r) f'(\varphi)^2 \varphi'(r) + \alpha f'(\varphi) \left( 4 e^{B(r)} \left( \left( -1 + e^{B(r)} \right)^2 + 4 r^2 \varphi'(r)^2 \right) + r A'(r) \right.
$$

$$
\left. \left( 4 \varphi'(r)^2 \left( e^{B(r)} r^2 + 2 \left( -1 + e^{B(r)} \right) \alpha f''(\varphi) \right) - 5 e^{B(r)} \left( -1 + e^{B(r)} \right) \right) \right) \right) r + e^{B(r)}
$$

$$
\alpha A'(r)^3 f'(\varphi) \left( r - 4\alpha f'(\varphi) \varphi'(r) \right) \left( e^{B(r)} r + 2 \left( -3 + e^{B(r)} \right) \alpha f'(\varphi) \varphi'(r) \right) + e^{B(r)} A'(r)
$$

$$
\left( e^{B(r)} \varphi'(r) \left( \left( e^{B(r)} r^2 - 4\alpha f''(\varphi) \right) \varphi'(r)^2 + 2 e^{B(r)} \left( -4 + e^{B(r)} \right) \right) r^3 - 12 \left( 15 - 8 e^{B(r)} \right.
$$

$$
+ e^{2B(r)} \right) \alpha^2 f'(\varphi)^2 \varphi'(r)^3 r + 4\alpha f'(\varphi) \left( -r^2 \left( e^{B(r)} r^2 + 4 \left( -2 + e^{B(r)} \right) \alpha f''(\varphi) \right) \varphi'(r)^4 \right.
$$

$$
\left. \left. - 2 \left( 3 e^{B(r)} \left( -3 + e^{B(r)} \right) r^2 + \left( 1 - 3 e^{B(r)} + 2 e^{2B(r)} \right) \alpha f''(\varphi) \right) \varphi'(r)^2 + e^{B(r)} \left( -1 + e^{B(r)} \right)^2 \right) \right)
$$

$$
- A'(r)^2 \left( e^{3B(r)} \varphi'(r) r^4 + 2 e^{B(r)} \alpha f'(\varphi) \left( \left( e^{B(r)} \left( -4 + e^{B(r)} \right) r^2 - 2 \left( -5 + 3 e^{B(r)} \right) \right.
$$

$$
\alpha f''(\varphi) \right) \varphi'(r)^2 + e^{B(r)} \left( -1 + e^{B(r)} \right) \right) r + 4\alpha^2 f'(\varphi)^2 \varphi'(r) \left( e^{B(r)} \left( 3 - 4 e^{B(r)} + e^{2B(r)} \right) \right.
$$

$$
\left. \left. - 2 \varphi'(r)^2 \left( e^{B(r)} \left( -3 + 2 e^{B(r)} \right) r^2 + \left( 9 - 8 e^{B(r)} + 3 e^{2B(r)} \right) \alpha f''(\varphi) \right) \right) \right) \Big) \Big/ \Big( -4 e^{2B(r)}
$$

$$
\left( e^{B(r)} r - 4\alpha f'(\varphi) \varphi'(r) \right) \left( r^2 \varphi'(r)^2 + e^{B(r)} - 1 \right) r^2 + 4 e^{2B(r)} m^2 \varphi(r)^2
$$

$$
\left( e^{B(r)} r^2 \left( e^{B(r)} r - 4\alpha f'(\varphi) \varphi'(r) \right) - 4 \left( -1 + e^{B(r)} \right) \alpha^2 A'(r) f'(\varphi)^2 \right) r^2 - 8 \left( -1 + e^{B(r)} \right)
$$

$$
\alpha^2 A'(r)^2 f'(\varphi)^2 \left( e^{B(r)} r + \left( -9 + 5 e^{B(r)} \right) \alpha f'(\varphi) \varphi'(r) \right) + e^{B(r)} A'(r) \left( 3 e^{2B(r)} r^4 \right.
$$

$$
+ 8 e^{B(r)} \left( -4 + e^{B(r)} \right) \alpha f'(\varphi) \varphi'(r) r^3 + 4\alpha^2 f'(\varphi)^2 \left( 5 \left( -1 + e^{B(r)} \right)^2 \right.
$$

$$
\left. \left. - 4 \left( -4 + e^{B(r)} \right) r^2 \varphi'(r)^2 \right) \right) \Big)
$$

(50)

875    with $e^{B(r)}$ given by (35).

### A.0.3   Near-horizon expansion in $1/A'$

877 In the near horizon limit, the expansion of the second order differential equations (36) in
878 terms of $1/A'(r)$ resulted in (39) with the coefficients given by

$$
\begin{aligned}
a =& -6\alpha^2 m^4 r^4 \varphi(r)^4 f'(\varphi)^2 - 2\alpha m^2 r^2 \varphi(r) f'(\varphi) \left( 2\alpha f'(\varphi) \varphi'(r) + r \right)^2 - m^2 \varphi(r)^2 \\
& \left( 4\alpha r^5 f'(\varphi) \varphi'(r) + 4\alpha^2 r^2 f'(\varphi)^2 \left( r^2 \varphi'(r)^2 - 4 \right) - 16\alpha^4 f'(\varphi)^4 \varphi'(r)^2 \right. \\
& \left. - 16\alpha^3 r f'(\varphi)^3 \varphi'(r) + r^6 \right) + 4\alpha r^3 f'(\varphi) \varphi'(r) + 2\alpha^2 f'(\varphi)^2 \left( 2 r^2 \varphi'(r)^2 - 3 \right) + r^4, \\
b =& \left( m^2 r^2 \varphi(r)^2 - 1 \right) \left( 4\alpha^2 f'(\varphi)^2 \left( 2 m^2 r^2 \varphi(r)^2 - 3 \right) - 8\alpha^3 m^2 r \varphi(r)^2 f'(\varphi)^3 \varphi'(r) \right. \\
& \left. + 2\alpha r^3 f'(\varphi) \varphi'(r) + r^4 \right), \\
c =& \alpha m^4 r^3 \varphi(r)^4 f'(\varphi) \left( r - 4\alpha f'(\varphi) \varphi'(r) \right) - m^2 r^2 \varphi(r) \left( 2\alpha f'(\varphi) \varphi'(r) + r \right)^2 \\
& - m^2 \varphi(r)^2 \left( 2\alpha r^2 f'(\varphi) \left( r^2 \varphi'(r)^2 + 1 \right) - 8\alpha^3 f'(\varphi)^3 \varphi'(r)^2 - 12\alpha^2 r f'(\varphi)^2 \varphi'(r) \right. \\
& \left. + r^5 \varphi'(r) \right) + \alpha f'(\varphi) \left( 2 r^2 \varphi'(r)^2 + 3 \right) + r^3 \varphi'(r).
\end{aligned}
$$

(51)

### A.0.4 Regularity condition

Requiring regularity of the scalar field at the horizon lead to (40) for the derivative of the scalar field at the horizon with coefficients

$$
\begin{aligned}
A &= -4\alpha^2 m^4 r_h^3 \varphi_h^4 f'(\varphi_h)^2 - m^2 r_h \varphi_h^2 \left(r_h^4 - 12\alpha^2 f'(\varphi_h)^2\right) - 4\alpha m^2 r_h^3 \varphi_h f'(\varphi_h) + r_h^3 \\
B &= 4\alpha f'(\varphi_h)\left(-m^2 \varphi_h^2\left(r_h^4 - 4\alpha^2 f'(\varphi_h)^2\right) - 2\alpha m^2 r_h^2 \varphi_h f'(\varphi_h) + r_h^2\right),
\end{aligned}
\tag{52}
$$

$$
\begin{aligned}
C &= 16\alpha^4 m^2 \varphi_h^2 f'(\varphi_h)^4\left(m^2 r_h^2 \varphi_h^2 - 6\right) + 48\alpha^3 m^2 r_h^2 \varphi_h f'(\varphi_h)^3 \\
&\quad + 8\alpha^2 r_h^2 f'(\varphi_h)^2\left(2m^2 r_h^2 \varphi_h^2 - 3\right) + r_h^6.
\end{aligned}
\tag{53}
$$

## B Theoretical arguments for a monotonically decreasing linearized scalar profile

With similar arguments as in the discussion in [64], one can deduce that the solution to (21) has to be a monotonically increasing function in terms of $u$ or decreasing in terms of $r$. We start from the linear-in-coupling equation of motion (21) hence work in the dimensionless parameter $u$ defined in (16). We are searching for solutions with a finite behavior at the horizon $u = 1$ and an asymptotically flat solution at infinity $u = 0$ as found in (28). This involves the following considerations:

*1) Once the solution becomes negative it can only become more negative, and cannot increase to zero again*

Suppose that the solution for the scalar profile becomes negative. To change sign again to positive values requires the existence of a minimum at negative field values. Multiplying (21) by $(-1)$ leads to

$$
(1-u)\varphi^{1''} - \varphi^{1'} = \frac{\hat{m}^2 \varphi^1}{u^4} - 3f'(\varphi^0)u^2 .
\tag{54}
$$

If there is an extremum for negative $\varphi^1$, we have $\varphi^1 < 0$ and $\varphi^{1'} = 0$ there. Hence at this location in between the boundaries

$$
(1-u)\varphi^{1''} = \frac{\hat{m}^2 \varphi^1}{u^4} - 3f'(\varphi^0)u^2 ,
\tag{55}
$$

Now the right hand side is $< 0$ and thus $\varphi^{1''} < 0$, since $(1-u) \geq 0$. Therefore, if there is an extremum for negative $\varphi^1$ it has to be a (local) maximum. This implies that field can only become more negative, which is incompatible with the required asymptotic behavior. Thus, for a positive coupling to have a solution that falls off to zero, the scalar field has to stay positive.

*2) The positive scalar field cannot have an extremum*

Next, we consider the case where the scalar field starts out positive. At a local maximum for a positive scalar field we have $\varphi^1 > 0$ and $\varphi^{1'} = 0$. Evaluating (21) at this location results again in (55). For a local maximum the second derivative should be negative, hence the right hand side should be negative as well. This implies the following inequality at the maximum

$$
\frac{\hat{m}^2 \varphi^1}{u^4} < 3f'(\varphi^0)u^2 .
\tag{56}
$$

Moving towards the horizon at $u = 1$ after a local maximum means $\varphi^1$ decreases and $u$ increases. Therefore the left hand side of the inequality (56) decreases and the right hand

911 side increases so the inequality holds. There cannot be an minimum because in that case (56)
912 would need to flip. Thus, the inequality holds up to the horizon. This further implies that the
913 slope of the profile at the horizon is negative or zero. However the differential equation at the
914 horizon is

$$-\varphi^{1\prime} = \hat{m}^2 \varphi^1 - 3f'(\varphi^0) \,. \tag{57}$$

915 Because the inequality (56) still holds at the horizon, the right hand side of (57) is negative.
916 This implies from (57) a nonzero positive derivative at the horizon, which is in contradiction
917 with the consequences of the inequality discussed above. Therefore, there cannot be a local
918 maximum for positive field values.
919 If the solution had a local minimum for the positive scalar field, it would require a local maxi-
920 mum as well to have an asymptotic fall off to 0, which we just argued cannot be the case. This
921 means that having a local minimum would lead to a diverging solution at infinity.
922
923 *3) The derivative of the scalar field at the horizon needs to be positive (or negative when working*
924 *in r)*
925    From the arguments above, the scalar field at linear order in the coupling needs to be
926 positive and cannot have local maxima or minima. Therefore the derivative of the scalar field
927 at the horizon at $u = 1$ needs to be positive to be able to connect to zero at infinity, because
928 a negative derivative at $u = 1$ leads to a ever increasing (or partly constant) function going
929 inwards to infinity, never reaching zero.

## C  Numerical methods

931 In this appendix we describe in detail the two numerical methods used to obtain the pertur-
932 bative and full solutions discussed in Sec. 3 and 4. Additionally a discussion on numerical
933 precision tests is given.

### C.1  Bisection method

935 Firstly in section 3.1.2 we describe solving the scalar field equation at linear order in the cou-
936 pling. As described in this section the asymptotic limit of the solution for the linearized scalar
937 field has an exponentially growing and decreasing mode (28). If not obtaining the initial con-
938 dition for which this growing mode is exactly zero, there will always be a large radial distance
939 at which the growing mode takes over and the solution diverges. Therefore a slight numerical
940 inaccuracy already leads to a divergence. Obtaining the exact solution is hard, however ap-
941 proaching the right initial condition is relatively easy. In this section we describe how one can
942 approach the right initial condition which corresponds to the exponentially decaying solution.
943    The differential equation (21) is approached as an initial value problem, starting the inte-
944 gration at an infinitesimal distance from the horizon $u = 1 - 10^{-5}$. The initial conditions are
945 given by (24). For a fixed mass and coupling, we vary the constant $\varphi_h^1$ to find the solution that
946 has an asymptotically flat limit. We obtain this by first determining an interval of $\varphi_h^1$ for which
947 the asymptotic behavior switches from positive infinity to negative infinity. By decreasing this
948 interval, the estimate of $\varphi_h^1$ corresponding to an asymptotically flat solution improves. We
949 implement this through the following algorithm, for each choice of $\hat{m}$:

950 • Make an initial guess $\varphi_h^1$ obtained by extrapolating (28) with $\bar{\varphi}_\infty^1 = 0$ and computing
951    its value at the horizon.

952 • Check if the solution corresponding to this guess diverges to positive or negative infinity.

- Incrementally increase (decrease) $\varphi_h^1$ if the solution with the initial guess diverges negatively (positively) and check the divergence behavior at each step.

- When reaching a step for which the divergence flips sign, defining this value as $\varphi_{h,flip}^1$, calculate $\varphi_{h,new}^1 = (\varphi_{h,initial}^1 + \varphi_{h,flip}^1)/2$.

- Use this mean value $\varphi_{h,new}^1$ as the new initial guess, decrease the step size every iteration by one order of magnitude.

- Continue these iterations until the guess saturates, where more iterations result in more accurate solutions.

In Fig. 2 we show the solution of (21) running the bisection method described above different number of times. One can see that for more cycles, the diverging behavior happens for smaller $u$/larger $r$. In principle, extending to infinite cycles, one would obtain the actual decaying solution. However the estimation for $\varphi_h^1$ would only differ infinitesimally, hence it is accurate enough to cut of the number of cycles at a finite value. In our analysis in section 3.1.2 and 4 we execute 15 cycles. This means that the estimation for $\varphi_h^1$ differs with an order of magnitude of $10^{-14}$ from the estimate at 14 cycles. This estimate therefore has very high accuracy, however the main reason applying this many cycles is to push the diverging behavior relatively close to $u = 0$ without making the computational time too long. The linearized scalar field solution is substituted in the higher order field equations and therefore the diverging behavior works through in the solutions for the metric functions and higher order scalar field as well. Therefore to get an accurate perturbative solution for as largest range of $u$ as possible, around 15 cycles or more is advised.

## C.2   Shooting method

For numerically calculating the perturbative solution to the metric functions in section 3.2 and the full solution in section 4, we use the so called shooting method. In this section we describe in more detail what this method entails.

The shooting method can be used as a numerical method to solve differential equations with a boundary value problem. This is the case for the modified Einstein equations for which we constructed the behavior of the metric functions at the boundaries; the near horizon and asymptotic limits. An additional requirement is that the solution does not have the instable behavior with respect to the initial conditions as is the case for the scalar field equation described in the previous section.

The shooting method is based on reframing the problem as an initial value problem with variable initial conditions. One integrates outwards to obtain the solution of this initial value problem for different guesses of the initial condition and evaluates the solution at infinity until these values at infinity agree with the boundary condition in the asymptotic limit. To describe this in more detail let us describe this for the specific case of solving the $tt$ component at second order in the coupling for the metric function $\bar{B}^2$ as is done in section 3.2. The boundary conditions are given by (31) and (32), where we can vary the near horizon constant $\bar{A}_h^2$.

- Construct a function $f[\bar{A}_h^2, u]$ of the differential equation solver from the black hole horizon outwards to infinity, in this case for the $tt$ component of (47) with initial conditions at the horizon following (31) with $\bar{A}_h^2$ as variable.

- Define the function of the asymptotic limit $g[u]$ as in (32) .

- Then define $h[\bar{A}_h^2] = f[\bar{A}_h^2, 0] - g[0]$ the difference between the solution to the initial value problem and the asymptotic limit evaluated at infinity $u = 0$.

- Find the root(s) of $h$, the value of $\bar{A}_h^2$ corresponding to the root is the correct initial condition to the boundary value problem and substituting this value for $\bar{A}_h^2$ in $f$ gives you the correct solution for $\bar{B}^2(u)$.

In our case the outer boundary lies in the asymptotic limit, however as one substitutes the linearized scalar field solution in the differential equations at higher orders in the coupling, the divergence behavior at finite $u$ of this scalar field also works through in the higher order equations. Therefore in practice instead of evaluating function $h$ at infinity, evaluate the functions at smallest possible $u$ before the divergence in $\varphi^1$ starts. This does slightly deteriorate the accuracy of the perturbative solution.

Following this calculation results in the following solution for $\bar{B}^2$ for a mass of $\hat{m} = 1$.

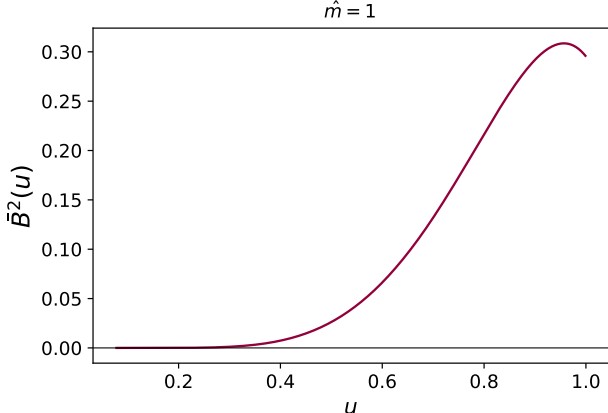

Figure 10: *Solution for $\bar{B}^2$ with $\hat{m} = 1$*. The shooting method resulted in $\bar{A}_h^2 = 0.295$ as initial condition in (31).

## C.3 Numerical precision tests

In this section we describe in more detail the used numerical method for the numerical integrator mentioned in Sec. 3, 4. We used the MATHEMATICA numerical integrator *NDSolve* for solving the boundary value problems in the perturbative and exact contexts. To obtain the numerical solutions for the metric functions and scalar field on itself the standard machine precision and "*StiffnessSwitching*" method in the *NDSolve* environment are sufficient, and no problems arise for the solutions and its derivatives. However, we encountered problems with numerical stability of the solutions in follow up calculations, more specifically when computing the percent difference of the Kretschmann scalar in Fig. 5. This arose as oscillatory behaviour of the final numerical function describing this percent difference. We therefore set up a (non-exhaustive) sweep over the different methods and working precision for the *NDSolve* environment to conclude which setting could mitigate this effect. Working from left to right we checked the following configurations, shown in Table 2.

We executed the tests in the following manner. We set up a module function with the boundary value problem for the exact field equations as described in Sec. 4 that computes the solution for the metric function $A(r)$ and $\varphi(r)$, with the method, WorkingPrecision, AccuracyGoal and PrecisionGoal as variables. In the same module we compute the percentual difference of the Kretschmann scalar in massive sGB substituting the solutions, with the Schwarzschild curvature invariant. From random test we had already found the oscillations due to limited numerical precision to worsen for smaller choices of the coupling constant, hence we chose to do the tests for $\hat{\alpha} = 0.01$, and to keep the running time manageable, we choose a small

| Methods | WorkingPrecision | AccuracyGoal | PrecisionGoal |
|---|---|---|---|
| "Adams" | 5 | 20 | 20 |
| "BDF" | 15 | 25 | 25 |
| "ExplicitRungeKutta" | 25 | 30 | 30 |
| "ImplicitRungeKutta" | 30 | | |
| "SymplecticPartitionedRungeKutta" | 40 | | |
| "MethodOfLines" | 50 | | |
| "Extrapolation" | | | |
| "DoubleStep" | | | |
| "LocallyExact" | | | |
| "StiffnessSwitching" | | | |
| "Projection" | | | |
| "OrthogonalProjection" | | | |
| "IDA" | | | |
| "StiffnessSwitching", Method → {"ExplicitRungeKutta", Automatic} | | | |
| "TimeIntegration" → {"ExplicitRungeKutta", "DifferenceOrder" → 8} | | | |
| "TimeIntegration" → "ExplicitEuler" | | | |
| "PDEDiscretization" → {"MethodOfLines", "SpatialDiscretization" →{"TensorProductGrid", "MinPoints" → 1000}} | | | |
| "PDEDiscretization" → {"MethodOfLines", "SpatialDiscretization" → {"FiniteElement"}}} | | | |

Table 2: Working from left to right, the different settings for the *Method, WorkingPrecision, AccuracyGoal* and *PrecisionGoal* within the *NDSolve* function, for finding the configuration mitigating the effect of numerical inaccuracy.

scalar field mass $\hat{m} = 0.01$. Additionally from the sample tests we found that some of the methods in Table 2 that did improve on the numerical inprecision issues, did not give output for the default WorkingPrecision, therefore in general we set the WorkingPrecision and MachinePrecision to **30**. First we computed the percent difference function up to $r/r_H = 10$ for the different methods in the first column of Table 2 with the AccuracyGoal and PrecisionGoal on default. We selected the method for which the function did not diverge at the horizon and which mitigated the oscillation the most, which resulted in *"TimeIntegration"* → *{"ExplicitRungeKutta", "DifferenceOrder"* → *8}*.

Then we repeated the calculation specifying to this method, now varying the WorkingPrecision found in the second column of Table 2. For precision below **WorkingPrecision = 25** in combination with above chosen method, the correct solution for the boundary value problem is not found, minimal precision of **WorkingPrecision = 25** is required. The oscillations got damped for higher values of the precision as expected, from **WorkingPrecision = 30** and onwards the oscillations up to $r/r_H = 10$ are smoothed out completely.

Lastly we repeated the computation for the above mentioned method and **WorkingPrecision = 25** for different values of the AccuracyGoald and PrecisionGoals given in the last two columns of Table 2, choosing values comparable to the set WorkingPrecision. Both tested separate from each other and in the different combinations, checking what configuration of these settings mitigated the oscillatory behaviour that is still present at this WorkingPrecision. We found no observable improvement on the oscillatory behavior from these two settings. Hence specifying **WorkingPrecision = 30** on itself results in sufficient numerical precision. The default setting for the AccuracyGoal and PrecisionGoal are both set as half the WorkingPrecision. For larger distances than $r/r_H = 10$ the precision still might be too limited but in principle one could solve this issue by improving on the precision settings. Note we also did not explore every permutation of settings, however for our purposes computing the solutions with method *"TimeIntegration"* → *{"ExplicitRungeKutta", "DifferenceOrder"* → *8}*, **WorkingPrecision = 30** and AccuracyGoal, PrecisionGoal on default, suffices.

# D   Additional analysis of perturbative solutions

In addition to the analysis in Sec. 5 we discuss in this appendix the perturbative solution in more detail, comparing the solution up to different orders in the coupling with the exact numerical case. The difference between the solutions is most noticeable in the near horizon region, where the spacetime curvature, see Fig. 5, is strongest and the scalar field energy density the highest, see Fig. 6.

Starting with the metric function $\bar{A}$ as defined in (18) with the perturbative solution rescaled to variable $r$ with the method described in Sec. 5. The top and bottom panel of Fig. 11 show the metric function near the horizon for $\hat{\alpha} = 0.2$ and $\hat{m} = 0.01$ and $\hat{m} = 0.1$ respectively. We zoom in on the region near the horizon as there the differences between the curves is most noticeable, for larger radial distances the curves coincide in all cases below as expected. In both panels of Fig. 11 one can see that the perturbative curves lie below the exact solution and above the Schwarzschild solution. Including corrections to higher order in the coupling for the perturbative solution results in the curve lying slightly closer to the exact solution as one would expect. The roots of the curves correspond to the respective horizon radii. Similar as we showed in 3 the horizon radius for the exact solution is smaller than the perturbative and Schwarzschild horizons. Furthermore from both panels of Fig. 11 we find the horizon radius shifts towards the horizon of the exact curve for higher corrections to the perturbative solution.

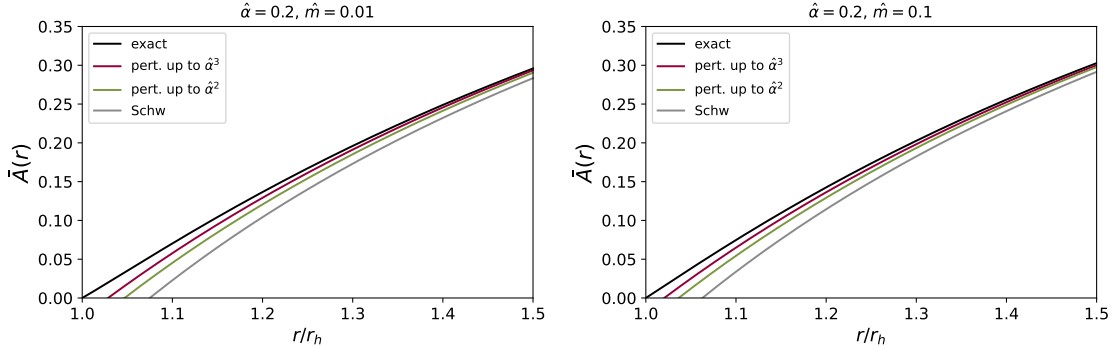

Figure 11: *The solution of metric function* $\bar{A}$ *close to the horizon, comparing the exact, perturbative solution up to* $\hat{\alpha}^3$, *up to* $\hat{\alpha}^2$ *and the Schwarzschild solution respectively.*

In Fig. 12 we show the perturbative solution of the scalar field up to linear, quadratic and cubic order in the coupling compared to the exact solution for $\hat{\alpha} = 0.2$ and $\hat{m} = 0.01$ and $\hat{m} = 0.1$ respectively. In both panels we find that the perturbative solution approaches the

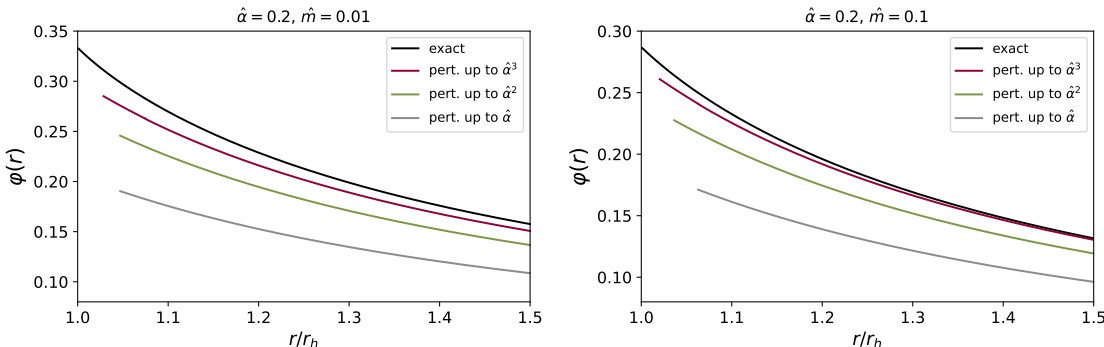

Figure 12: *The solution of* $\varphi$ *close to the horizon, comparing the exact, perturbative solution up to* $\hat{\alpha}^3$, *up to* $\hat{\alpha}^2$ *and the linearized solution respectively.*

exact solution from below and again becomes more accurate with increasing orders in $\hat{\alpha}$, as one would expect. Furthermore, comparing the top and bottom panel we find the an increased accuracy of the perturbative solution for larger scalar field mass where the improvement is more noticeable than for the metric function in Fig. 11. For the case of the scalar field we do not find any particular change comparing the second order solution (green curves) to the cubic order solution (pink curves). This is interesting as for the latter, the corrections to the metric function first contribute to the scalar field solution, see Table 1.

From the analysis in this appendix together with Sec. 5 we can conclude that the perturbative solution becomes more accurate for small values of the coupling, large values of the scalar field mass and/or large distances from the horizon. The comparison does not show any qualitatively new non-perturbative behaviour that would not be captured by the perturbative solution when adding higher order corrections to increase accuracy.

# E  Calculation of the ISCO and light ring radii

In this appendix we show how one can determine the ISCO radius and light ring radius in Schwarzschild coordinates and the corresponding orbital frequencies. This is used in Sec. 6.3.1. The ISCO radius can be determined from the effective potential. Starting from a static spheri-

cally symmetric metric (11), one can write down the normalization of the four velocity $g_{\mu\nu}\dot{x}^\mu\dot{x}^\nu = -1$. Before writing this down explicitly we can use the symmetries of the spacetime, e.g. as the metric components are independent of $\phi$ and $t$ there are two constants of motion $E = -e^{A(r)}\dot{t}$ and $L = r^2\dot{\phi}$, the energy and angular momentum per unit mass. As the conservation of (the direction of) angular momentum requires the motion of a particle to be planar, together with rotational symmetry, one can fix the motion to be equatorial with $\theta = \frac{\pi}{2}$. Substituting these quantities in the normalization condition we obtain

$$
e^{B(r)}\dot{r}^2 = -1 + e^{-A(r)}E^2 - \frac{L^2}{r^2}\,,
$$
$$
\dot{r}^2 = V_{eff}(r)\,,
\tag{58}
$$

with

$$
V_{eff}(r) = e^{-B(r)}(-1 + e^{-A(r)}E^2 - \frac{L^2}{r^2})\,,
\tag{59}
$$

the effective potential as (58) now describes the equation for a classical particle moving in potential $V_{eff}(r)$[5]. Additionally to the effective potential we can write down the radial component of the geodesic equation

$$
\dot{r}^2 + \frac{e^{A(r)'}}{2e^{B(r)}}\dot{t}^2 - \frac{r^{2'}}{2e^{B(r)}}\dot{\phi}^2 = 0\,.
\tag{60}
$$

For finding the innermost stable circular orbit we are interested in circular orbits and therefore $\dot{r} = \ddot{r} = 0$. Substituting these conditions in (60) and using this equation to construct the angular frequency $\omega = \frac{\dot{\phi}}{\dot{t}}$ results in

$$
\omega^2 = \frac{e^{A(r)'}}{r^{2'}}\,.
\tag{61}
$$

Then combining (58), the condition for circular orbits, the definitions of the constants of motion and (61) we find for the energy and angular momentum per unit mass for circular orbits

$$
E = -\frac{-e^{A(r)}}{\sqrt{e^{A(r)} - r^2\omega^2}}\,,
$$
$$
L = \frac{r^2\omega}{\sqrt{e^{A(r)} - r^2\omega^2}}\,.
\tag{62}
$$

Now $V_{eff}(r)$ has two extrema, the inner extremum is a maximum corresponding to an unstable circular orbit and the outer with a minimum an thus a stable circular orbit. The minimum radius for this stable circular orbits happens when these two extrema coincide, this is when the second order derivative of $V_{eff}(r)$ has a root. Therefore taking the second order derivative to $r$ of (59) treating $E$ and $L$ as constants of motion, followed by substituting (62) and finding the radius that corresponds to the root results in $r_{ISCO}$. Substituting this radial coordinate in (61), one obtains the orbital frequency at the ISCO radius, which is in contrary to the radius a coordinate independent quantity.

---

[5]In the literature there are slightly different interpretations of $V_{eff}$ e.g. sometimes the $E^2$ term is treated separately or the sign might be opposite. The definition of (59) intuitively makes sense as for large $L$ there are two extrema which correspond to circular orbits. The extremum closest to the horizon correspond to a maximum and hence the unstable orbit and the outer extremum to a minimum, the stable orbit. The innermost stable orbit is found at the point where the two extrema coincide, hence for this purpose the different interpretations generally do not matter. However the interpretation of the extrema in this way of defining the effective potential makes most sense.

Finding the light ring is a bit more straight forward. Photons travel along null paths $ds = 0$, additionally we are interested in circular orbits $dr = 0$ and for similar arguments as before we can set $\theta = \pi/2$ hence $d\theta = 0$. This simplifies the equation for null paths to

$$\dot{\phi}^2 = \frac{e^{A(r)}}{r^2} \, . \tag{63}$$

Additionally from the radial component of the geodesic equation in circular orbits we obtained (61), substituting this in (63) gives

$$\frac{e^{A(r)\prime}}{r^{2\prime}} = \frac{e^{A(r)}}{r^2} \, , \tag{64}$$

solving for $r$ results in $r_{LR}$. Substituting this radial coordinate in (61), one obtains the orbital frequency at the light ring radius.

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
