# Peer review of "Massive scalar clouds and black hole spacetimes in Gauss-Bonnet gravity"

_SciPost Physics Core_

## Round 1 · Referee Report · Anonymous (Referee 1) · 2024-7-24

Report

The paper explores black holes in Einstein-dilaton-Gauss-Bonnet gravity with a massive scalar field. Various properties of the black holes are calculated and a comparison between perturbative and non-perturbative black hole solutions is performed. Constraints on the theory parameters are deduced based on the observed minimum and maximum black hole masses. The paper is very detailed, represents an interesting work, and investigates certain features of the Gauss-Bonnet black holes that were previously unexplored. There are certain points, though, that require further clarification or polishing:

1. Constraints on the Gauss-Bonnet theory parameters, for the specific flavor of the theory employed by the authors, are considered also in other papers. For example, the constraints coming from the existence of a minimum mass neutron star can be found in arXiv: 1109.0928,2110.02997,1510.02152,2402.06305 (minimum mass black hole mass constraints are also connected in some of these papers). Moreover, the binary pulsar observations at least for now give stronger constraints (connected to their maximum mass and scalar dipole radiation) compared to the binary merger observations. The authors should compare their results to these previous findings.
2. You say that “we conclude that the perturbative solution also becomes more accurate in the large scalar mass regime”. I am not sure this observation is accurately expressed. Large scalar field mass decreases the black hole scalar field. For me, the better accuracy is an effect of the smaller scalar field that effectively reduces the coupling.
3. Apart from the comparison of single solutions, as in Fig. 3, I think that a comparison between whole sequences of black holes should be visualized (for different scalar field masses). This will make the outline and conclusions in the paper easier to follow. I think this is especially important for Sec. 6.2.1. – it is a bit difficult to imagine the origin of constraints without understanding how the spectrum of solutions behaves (e.g. in a M(rh) diagram)
4. The proof of the requirement phi’(rh)>0 seems to be done only in the perturbative regime, i.e. for weak coupling. On the other hand, the authors demonstrate that for strong coupling the perturbative solution is not accurate (e.g. in Fig. 3). Therefore, I am not sure that this should be used as a general requirement for imposing constraints. Perhaps the authors should justify this point a little bit better.
5. Fig. 7 - I do not understand well the discrepancy between the green and the black dots at certain places. In the right-upper figure, the black square sequence has a “bump” slightly after m=1, while this “bump” is not present in the right-lower panel (in that panel also the black and green dots at least visually coincide). What is the origin of this bump and why is a difference only seen in the right-upper panel?
6. It will also be nice to comment on the dependence of your results on the free parameter in the coupling function exponent.
7. The conclusions in Section 6.2.1. are a bit difficult to follow in my opinion. Since it is one of the main parts of the paper, please explain your arguments in more detail.

Recommendation

Ask for major revision

  • validity: -
  • significance: -
  • originality: -
  • clarity: -
  • formatting: -
  • grammar: -

Author:  Iris van Gemeren  on 2024-09-20  [id 4791]

(in reply to Report 1 on 2024-07-24)
Category:
answer to question

We would like to thank the referee for the comments and evaluation of the manuscript. We reply to the referee’s specific comments point by point below.

1)
We agree with the referee that our discussion of the literature on the constraints on the coupling of sGB is limited. We only cited previous work on the strongest constraints coming from BH binary mergers whereas there are constraints on shift symmetric/dilatonic sGB in the weak field limit from a great variety of sources also including solar system test and the above mentioned neutron star solutions and binary pulsar observations. Therefore we decided to mention and cite the broader variety of literature on the topic in the resubmission file, including among others the papers highlighted by the referee. However we did decide not to make a separate section on the constraints of the theory as nice overviews are already present in other work (e.g. table 1 in 2201.02543 and 2110.02997). So far we were not able to find previous work containing stronger constraints on the coupling from binary pulsar observations compared to black hole binary results. We do find comparable constraints (2208.09488) and stronger constraints for larger exponents of the exponential in the coupling function for dilatonic sGB theory (2402.06305), therefore we agree with the importance of including these previous works however we state the strongest constraints on the massless theory are the result of the already cited BH binary merger observations.

2)
We agree with this rectification of the referee. The perturbative solution indeed does not per se become more accurate, this would be the case if, next to the expansion in the small dimensionless coupling, we also expanded in the inverse of the dimensionless scalar field mass, which we do not. We therefore rephrased this statement in the manuscript for resubmission by noting that the difference with the exact solution decreases for larger scalar masses due to the decreased scalar field in this regime, removing the accuracy statement.

3)
Initially we did not include an overview of the spectrum of solutions in a M(r_h) diagram as it is already present in fig.5 of 1903.08119. Although we agree that for the sake of completeness it would be nice to add and the interpretation of our results compared to 1903.08119 is not one-to-one as in the aforementioned paper, the parameters are rescaled by the coupling instead of the horizon radius. We decided therefore to include a diagram showing the fractional difference of the horizon radius to the Schwarzschild radius as function of the black hole mass in solar masses in the manuscript for resubmission. Showing different curves varying the scalar field mass and the coupling constant, seeing the effect of both on the relation between the mass and the horizon radius. This adds additional information to fig.5 of 1903.08119 seeing the effect of both parameters instead of the combined $\sqrt(\alpha) m$ dependency. The diagram shows the offset of the horizon radius compared to the Schwarzschild radius and the minimum mass of the black holes for the different parameter choices. As the minimum black hole mass is linked to preventing the finite radius singularity to lie outside the horizon we come back to this diagram in Sec. 6.2.1. as suggested by the referee, where we discuss the maximum dimensionless coupling as a function of the scalar field mass which we showed is also related to censoring the singularity behind the horizon.

4)
As stated, we show explicitly that the linearized scalar field solution is a monotonically decreasing function in terms of parameter $r$ and hence has a negative derivative at the horizon in Appendix B. We intend to add the following explanation on why we assume $\varphi’(r_h)<0$ for general coupling as well: ‘For the scalar field background solution for general coupling we expect to find ground state type behavior as well, corresponding to a monotonically decreasing solution, as sudden bumps or wells are expected for excited states. This assumption is also built upon the fact that the full exact solution should reduce to the linearized solution in the small coupling limit. Also numerically, the solutions to eq. (36) found to connect the near horizon and asymptotic limits of the scalar field for generic coupling, see Fig.4, shows this type of function too. Hence we state the scalar field at the horizon for generic coupling ought to be negative to be able to find a solution to eq. (36) connecting the near horizon and asymptotic limits.’

5)
The bump in the right upper figure has the same origin as the cusp feature shown in the dashed curves of the left panel of Fig.7. These cusp points correspond to a coupling dependent bifurcation point from which two branches of value for phi_h arise for which $\varphi’(r_h)<0$. For the top branch $\varphi_h$ grows with the scalar mass and for the bottom branch it decreases. We expect the scalar field to decrease and finally decouple in the large scalar mass limit (this is also what the values for $\varphi_h$ show corresponding to the actual numerical solution, the dots and diamonds in the left panel of Fig.7), hence we only show the bottom branch of $\varphi_h$ values corresponding to the near horizon constraint. Same we do for the values for the maximized $\varphi_h$ also corresponding to the near horizon constraint in the upper right panel of Fig.7. As this feature is not visible in the panel for the max coupling we find that the coupling is not very sensitive to the initial condition of the scalar field.
To make the argument and procedure of how the right panels of Fig.7 are produced a bit clearer we intend to only keep the bottom panel on the maximum coupling in the main discussion. As the cusp feature is already visible in the left panel and our main discussion on comparing two types of constraints is also captured by showing only the bottom right panel. The top right panel of the maximum $\varphi_h$ is then placed in an additional appendix E with a step by step description on how the values of $\alpha_{max}$ and $\varphi_{h, max}$ are obtained.

6)
In the first paragraph of Sec.5 we intend to add a few sentences on the free parameter in the exponent: ‘As mentioned in Sec.2.1 we focus on $\gamma=2$ as parameter in the exponent in the coupling function corresponding to EdGB. We fix this choice to keep the amount of free parameters tractable. Additionally, the effect on the properties of the BH solution because of $\gamma$ was studied in 1903.08119, based on these results one finds qualitatively similar behavior for other choices of this parameter. In general the effect of larger values of $\gamma$ correspond to an enlarged effect from the scalar field leading for example to stronger constraints on the coupling see 2402.06305.’

7)
In Sec. 6.2.1 we first compare the constraint on the coupling constant for massive sGB from previous work to the theoretical constraint we find in the right panel of Fig.7. The constraint from 1905.11859 of $\sqrt(\alpha)<= 2.47$ km was found for a scalar mass range of $10^{-15} eV<=m<=10^{-13} eV$. To compare this to our result we first needed to convert this massrange to the values of our dimensionless mass $\hat{m} = (m_{eV} r_h e)/(\hbar c)$. We intend to explain this more elaborately in a step by step manner obtaining this equation for the conversion of the mass. As a proxy for $r_h$ we used the Schwarzschild radius for black holes in the stellar BH mass range as the aforementioned constraint was based on LVK observations. Substituting the mass range in eV and the radii in the formula for $\hat{m} $ we obtained that the constraint of $\sqrt(\alpha)<= 2.47$ km holds for $ 10^{-5}<=\hat{m}<=10^{-1}$. As in the right panel of Fig.7 our theory constraint on the coupling is shown for masses larger than $10^{-1}$ the constraint shown in this figure is the first one for larger scalar field masses.

Lastly in this section, we wanted to give some more intuition on the magnitude of our dimensionless parameter $\hat{m}$ to convert it to its dimensionfull counterpart and compare it to scalar particle masses. We picked therefore a representative value for $\hat{m}$ focussing on $\hat{m} \sim 1$ as being the largest possible mass for which there is still a significant scalar field, as the scalar field is strongly suppressed for larger masses. Converting this value to the dimensionfull mass again with $ m_{eV} =(\hat{m} \hbar c)/ ( r_h e)$, this time not putting any restrictions on the type of black holes and taking for the proxy of the black hole horizon the full range of astrophysical black holes $5 M_{\odot} <= M <= 10^{10} M_{\odot}$. This resulted in the mass range $10^{-21} eV<=m<= 10^{-11} eV$ as an example of the magnitude of scalar field masses we studied with our choices of values of $\hat{m} \sim 1$. Comparing this to the mass ranges of ultralight dark matter models to place it into context we could conclude the calculated representative mass range lies within the mass ranges of these models.

---

## Editorial Decision

resubmitted